# Absolute Coordinates Make Motion Generation Easy

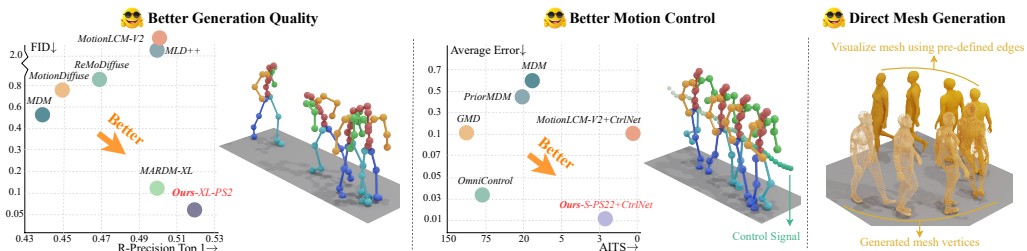

Figure 1: **Absolute coordinates make motion generation easy.** Here we show that our model produces motion of higher fidelity, has better controllability, and reports promising results of generating SMPL-H meshes directly.

## Abstract

State-of-the-art text-to-motion generation models rely on the kinematic-aware, local-relative motion representation popularized by HumanML3D, which encodes motion relative to the pelvis and to the previous frame with built-in redundancy. While this design simplifies training for earlier generation models, it introduces critical limitations for diffusion models and hinders applicability to downstream tasks. In this work, we revisit the motion representation and propose a radically simplified and long-abandoned alternative for text-to-motion generation: absolute joint coordinates in global space. Through systematic analysis of design choices, we show that this formulation achieves significantly higher motion fidelity, improved text alignment, and strong scalability, even with a simple Transformer backbone and no auxiliary kinematic-aware losses. Moreover, our formulation naturally supports downstream tasks such as text-driven motion control and temporal/spatial editing without additional task-specific reengineering and costly classifier guidance generation from control signals. Finally, we demonstrate promising generalization to directly generate SMPL-H mesh vertices in motion from text, laying a strong foundation for future research and motion-related applications.

## 1 Introduction

Generating realistic human motion from textual descriptions has rapidly emerged as a significant research area. It has great potential for diverse applications, including virtual and augmented reality experiences, immersive metaverse environments, video game development, and robotics.

Recently, the introduction of the large-scale HumanML3D [25] dataset has catalyzed significant progress in text-to-motion generation by establishing a standardized, kinematic-aware motion representation. Earlier methods based on AutoEncoders [44, 4], GANs [22], or RNNs [82] attempted to model joints and kinematic rotations or joint and trajectory [1, 78, 5, 109, 55, 19], but struggled to

produce high-fidelity motion. HumanML3D instead proposes to encode motion relative to the pelvis and to the previous frame, enabling explicit modeling of intra-frame kinematics and inter-frame transitions. This local-relative, kinematic-aware representation, combined with built-in redundancy (non-animatable features such as incorrectly processed [101] relative rotations, local velocities, and foot contacts) as a form of data-level regularization [66], substantially simplifies training [12, 62, 66] and boosts the performance of these simple backbones. Recent diffusion-based methods [91, 119, 43] also adopt this representation for text-to-motion generation tasks as default, yielding state-of-the-art performances. While later works have explored architectural improvements [10, 90, 125, 33, 122], generation speedups [12, 15, 14], and retrieval-based enhancements [120], the underlying representation has been largely inherited from HumanML3D [25] without much careful study.

However, this *de facto* representation introduces several fundamental limitations. First, although this representation benefits earlier methods, the redundancy makes it difficult for diffusion models to learn [66], often leading to underperformance in generated motion quality. Second, its inherently relative nature is misaligned with the requirements of downstream tasks such as motion control and temporal/spatial editing [102, 42, 77]. These tasks demand motion generation that is not only semantically meaningful but also aware of absolute joint locations, which are usually provided by users, to enable precise control and intuitive motion editing. Attempts to inject absolute location information into the existing local-relative representation have often resulted in overly complex designs [52] and degraded generation fidelity [42, 102, 15, 14].

In this paper, we revisit the foundational question of motion representation for text-driven motion generative models. We begin by demonstrating that the redundant, local-relative, kinematic-aware formulation—commonly assumed to be essential—is not crucial for the performance of diffusion-based models. Instead, we adopt a much simpler and long-abandoned non-kinematic representation in text-to-motion methods: absolute joint coordinates in global space. Through careful analysis of key design choices, we show that even with a simple Transformer [93] model (*e.g.* without UNet [33, 10] or altered attentions [10, 120]) and without additional kinematic losses, this simple formulation can achieve significantly higher motion fidelity, improved text alignment, and strong scalability potential.

Furthermore, we show this simple representation naturally supports a range of downstream tasks, including motion control and temporal/spatial editing, without requiring task-specific reengineering. With inherent absolute location awareness, our formulation enables direct controllability by eliminating the need for relative-to-absolute post-processing, which often introduces errors, as well as removing reliance on time-consuming classifier guidance from control signals during generation.

By discarding the constraints of redundant, local-relative, kinematic-aware representation designs, our approach also opens the door to directly modeling motion from textual inputs beyond standard human joint skeletons. Our formulation shows potential to generalize to other subclasses of absolute coordinates, such as SMPL-H mesh vertices [59] in motion from text, which are largely neglected by existing approaches but crucial toward having vivid, animatable human avatars. This lays a foundation for future research in broader text-to-motion generation domains, enabling new applications across diverse motion-related domains.

In summary, our contributions are as follows:

- We propose a new formulation for text-to-motion diffusion models using absolute joint coordinates. Through systematic analysis of design choices, our method can achieve state-of-the-art performance with simple Transformer [93] backbones and no auxiliary losses.
- We demonstrate that this formulation naturally supports downstream motion tasks, including motion control and temporal/spatial editing, achieving better performance and enabling seamless integration without additional reengineering or time-costly guidance generation.
- We further show promising generalizes beyond joints to directly modeling other subclasses of absolute coordinates, such as mesh vertices. This flexibility marks an important step toward text-driven motion generation across broader domains and serves as a foundation for future research and broader real-world applications.

## 2    Related Works

**Human Motion Generation.** Early approaches in text-driven motion generation [1, 25, 70, 71, 89, 110] attempt to align the latent spaces of text and motion. However, these methods faced significant challenges in generating high-fidelity motion due to the difficulty of seamlessly aligning

two fundamentally distinct modalities. Inspired by the success of denoising diffusion models in image generation [29, 86], several pioneering works [91, 43, 119, 12, 115] introduced diffusion-based approaches for human motion generation. Subsequent works have primarily focused on architectural innovations [3, 116, 129, 10, 33, 14, 90, 125, 122, 101] or on improved training methodologies [51, 116, 2, 31, 95, 129, 61, 15, 120]. Other human motion generation works introduce Vector Quantization (VQ), enabling discrete motion token modeling [26, 117, 114, 77, 23, 8, 76, 127, 50, 37, 121, 63, 123] or explore autoregressive generation [124, 11, 85, 125, 90, 66, 101]. Recent works also diversified their focus, exploring human-scene/object interactions [69, 32, 45, 105, 74, 48, 9, 97, 21, 112, 107, 64, 13, 100, 38, 46, 58, 17, 126, 96, 108, 60, 35, 106], human-human interaction [36, 104, 99, 20, 53, 7, 113], stylized human motion generation [128, 24, 49], more datasets [103, 56], long-motion generation [131, 72], shape-aware motion generation [92, 54], fine-grained text controlled generation [132, 34, 111, 84, 39, 81, 88], leveraging 2D data [40, 73, 47], as well as investigating advanced architectures [122, 98]. In contrast, our work revisits the underlying text-to-motion representation itself. We show that adopting a simpler yet long-abandoned alternative: absolute joint coordinates, even with a simple Transformer backbone and no additional constraints, can significantly improve generation quality.

**Controllable Text-to-Motion Generation.** In addition to synthesizing motion purely from text prompts, recent work has explored controlling motion generation with auxiliary signals such as trajectories or editing constraints [42, 102, 15, 14, 75, 80, 94, 41]. Early approaches such as Prior-MDM [83] extended MDM [91] to support end-effector constraints. GMD [42] introduced spatial control by guiding the diffusion process on the root joint trajectory, but required a re-engineered motion representation specifically designed for the task. OmniControl and MotionLCM [102, 15] generalized control to arbitrary joints by leveraging ControlNet [118], but both still rely on relative motion representations. Moreover, OmniControl heavily depends on classifier guidance from control signals during generation; without it, motion quality degrades significantly. Input optimization-based approaches [41, 75, 14] proposed directly optimizing the inputs to meet control objectives, but suffer from high computational and time costs due to multi-round optimization and gradient accumulations, making real-time applications impractical. In this work, we show that our proposed absolute joint coordinate formulation enables superior performance without the need for task-specific reengineering and time-consuming classifier guidance or inference-time optimization.

**Mesh-Level Text-Driven Human Motion Generation.** Previous works rarely perform direct mesh vertex generation. Instead, prior methods [91, 12, 102, 42] typically predict HumanML3D representations and convert them to joint positions, followed by SMPL fitting [6]. Other efforts in related fields such as Human-Object Interaction (HOI), Human-Scene Interaction (HSI) and Dual-Person motion generation have attempted to directly model SMPL parameters [32, 45, 21, 64, 46, 100, 38, 58, 126, 96, 104] or joint rotations and translations [74, 48, 112], which are then applied to meshes through standard skinning and rigging techniques. However, SMPL fitting is time-consuming and prone to reconstruction errors, while directly modeling SMPL parameters or joint transformations remains challenging and often results in unsatisfactory mesh quality[69, 46]. Moreover, even small joint-level errors can be magnified when propagated to mesh vertices, degrading the visual fidelity of the synthesized motion. Direct mesh vertex generation from textual inputs remains largely unexplored, yet it is critical for achieving high-fidelity, visually realistic motion synthesis. In this work, we show that with our absolute coordinate formulation, we can naturally extend to directly generating mesh vertices from text and achieve strong performance.

**Text-to-Human Motion Representation** Early text-to-motion generation methods, often based on AutoEncoders [44, 4] or GANs [22], attempted to directly predict absolute joint positions [1], but struggled to produce realistic motions. Later approaches incorporated human kinematics by predicting joint rotations [78, 5, 109], combining joint positions with trajectory modeling [55, 19]. However, these designs remained limited in producing high-fidelity and semantically aligned motions. The HumanML3D [25] representation addressed these challenges by encoding motion relative to the pelvis and the previous frame, explicitly modeling intra-frame kinematics and inter-frame transitions. Its local-relative, kinematic-aware design, with built-in redundancy [66, 101] from features such as relative rotations, local velocity, and foot contacts, substantially simplified training [12, 62] and quickly became the dominant choice for subsequent text-to-motion generation methodologies. In this work, we demystify the significance of HumanML3D representation formulation and adopt a simpler, long-abandoned non-kinematic formulation: absolute joint coordinates in global space. We show that, with this design, our method achieves better performance using simple Transformer [93]

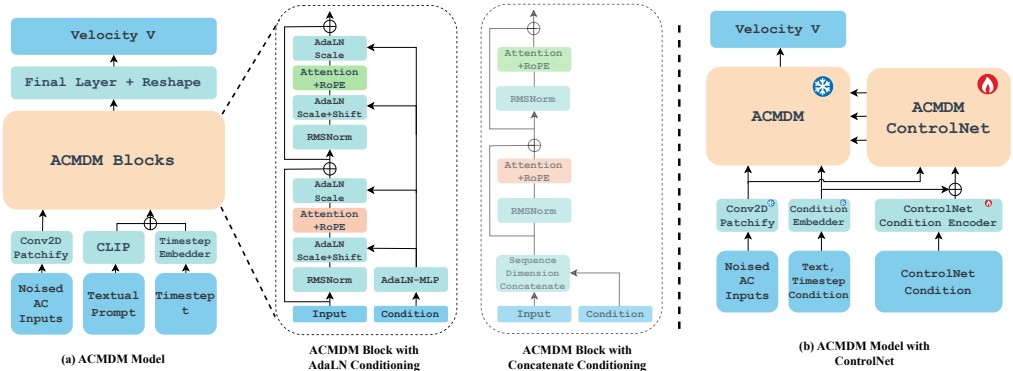

Figure 2: **Overview of our proposed ACMDM.** (a) Left: The raw/latent absolute coordinates representation is patchified and processed through a sequence of ACMDM blocks. Right: Details of ACMDM blocks, where we experiment with two conditioning variants: concatenation and AdaLN. (b) ControlNet-augmented ACMDM for controllable motion generation: Structured control signals are separately encoded and fused into the ACMDM generation process via additive residuals at each ACMDM block, enabling the model to follow both semantical and spatial controlling constraints.

backbones without auxiliary losses, and naturally extends to direct modeling other subclasses of absolute coordinates such as mesh vertices.

## 3  ACMDM: Absolute Coordinates Motion Diffusion Model

The majority of recent methods utilize the redundant, local-relative, and kinematic-aware motion representation popularized by HumanML3D [25]. However, this explicit inter-frame kinematic modeling around the pelvis makes the generation prone to accumulating global drift errors through frames, while the intra-frame relative formulation makes it difficult to incorporate absolute location controlling signals for downstream tasks. In contrast, we propose adopting a much simpler but long-abandoned alternative, absolute joint coordinates in global 3D space and show it makes human motion generation easy.

We first introduce our proposed ACMDM in Section 3.1 that we will systematically investigate and ablate in the experiments section. Next, in Section 3.2, we describe how to extend ACMDM to controllable motion generation through ControlNet integration without much task-specific engineering. Finally, we show how ACMDM generalizes to direct mesh vertex motion generation in Section 3.3.

### 3.1  Absolute Joint Coordinates for Text-to-Motion Diffusion

**Absolute Coordinates Representation.** We define absolute joint coordinates at each frame as $\mathbf{X}^i \in \mathbb{R}^{N_j \times 3}$, where $N_j$ is the number of joints (*e.g.*, 22 for the HumanML3D dataset), and each joint is represented by its 3D global position (XYZ). This intuitive formulation naturally avoids pelvis drift accumulation and facilitates direct controllability over spatial control signals. Previous works generally avoided this representation due to concerns about generating unnatural, non-human-like motions [102]. It was widely believed [102, 12, 62] that kinematic features were essential for physically plausible motion synthesis. In the experiment section, we demonstrate that using redundant kinematic features actually degrades motion generation quality, and that absolute joint positions alone are sufficient to achieve high-fidelity and controllable motion generation.

**Tokenizing Motion Representation.** Absolute joint coordinates inherently preserve both spatial and temporal structure of the motion data. Given a motion sequence input of shape $(L, N_j, d_{\text{in}})$, where $L$ is the motion sequence length and $d_{\text{in}}$ is the input feature dimension (3 for raw absolute coordinates), we apply a 2D convolutional layer to transform this structured input into a sequence of $T$ tokens similar to ViT [18], each with hidden dimension $d$. The number of tokens $T$ is determined by the predefined patch size $(P_T, P_S)$, where the convolution kernel size and stride are both set equal to $(P_T, P_S)$, resulting in $T = \frac{L}{P_T} \times \frac{N_j}{P_S}$. Importantly, we perform tokenization only along the spatial

(joint) dimension while preserving the full temporal resolution (*i.e.*, we define $P_T = 1$), as temporal details are especially critical for motion modeling. In our design, we explore various patch sizes, including $1 \times 22$, $1 \times 11$, and $1 \times 2$, corresponding to different joint-wise granularities.

**Motion Diffusion with Transformer.** After tokenizing the absolute coordinate inputs, the resulting token sequence is fed directly into Transformer for diffusion-based motion generation. Note that the central goal of this work is not to advance model architecture for motion generation. Rather, we focus on investigating the absolute coordinates motion representation. Therefore, we simply adopt a simple Transformer similar to DiT [68] and found it works sufficiently well.

To incorporate conditioning signals, we follow prior works [91, 117, 23, 77, 66, 76, 12, 120] and use a pretrained CLIP-B/32 [79] text encoder to extract the textual embedding $c$, along with a timestep embedder to process diffusion timestep $t$. We explore two conditioning mechanisms within our ACMDM design: (1) Concatenation, a commonly used method in prior text-to-motion works [91, 117, 23, 77, 76, 12], where condition vectors are appended along the sequence dimension; and (2) AdaLN, where the text and timestep embeddings modulate each block via adaptive layer normalization, similar to image diffusion DiT [68]. An illustration of these variants are shown in Figure 2 (a). In line with recent best practices in Transformer models, we also adopt several modern architectural components: Rotary Positional Embedding (RoPE) [87] and QK Normalization [28] are applied in the attention layers, and SwiGLU activations[67] are used in the feed-forward networks (FFNs). We also investigate different denoising targets for training ACMDM, including predicting $\mathbf{x}_0$ [29] (the original motion), $\epsilon$ [29] (the added noise), and velocity [57] $\mathbf{v}$ (under flow-matching formulations). In our experimental analysis, we show that $\mathbf{v}$ prediction consistently yields the best generation performance. All ACMDM variants are trained with a standard $L_2$ reconstruction loss on the diffusion objective. More details are provided in the supplemental material.

After processing through the motion diffusion Transformer, the output token sequence is linearly projected to match the original shape. Specifically, a linear layer is applied to transform each token from dimension $d$ back to $d_{in} \times P_T \times P_S$. The output is then reshaped to recover the original 2D structure (*i.e.*, $(L, N_j, d_{\text{in}})$) of the absolute joint coordinates.

**Latent Motion Encoding with a Motion AutoEncoder.** Optionally, we convert raw absolute coordinates into latents using a motion autoencoder (AE) and perform motion diffusion then, which leads to better generation fidelity as shown in the experiment section. Specifically, given a motion sequence $\mathbf{X}^{0:N} \in \mathbb{R}^{L \times N_j \times 3}$, a 2D ResNet-based encoder compresses it into a latent representation $\mathbf{x}^{0:n} \in \mathbb{R}^{l \times N_j \times d_j}$, where $l$ denotes the downsampled motion sequence length and $d_j$ is the dimension of the motion latent. We keep the number of joints $N_j$ unchanged here. Tokenization is then performed over the latent representations (so $d_{in} = d_j$), whose output will be fed into the motion diffusion Transformer. A decoder later can reconstruct the motion sequence $\hat{\mathbf{X}}^{0:N} \in \mathbb{R}^{L \times N_j \times 3}$ via nearest-neighbor upsampling based on the diffusion output. We explore a causal AE (*i.e.*, convolution kernels can only access previous frames), a non-causal AE, a VAE-based variant, and direct modeling on raw absolute joint coordinates in the experimental section. All these motion AE variants are trained with a simple smooth $L_1$ reconstruction loss. More details of all the AE variants are provided in the supplemental material.

**Scaling ACMDM.** We scale the model capacity by increasing the motion diffusion Transformer layer's depth and width. Specifically, we follow a simple scaling strategy where the number of Transformer layers is set equal to the number of attention heads. We define four model sizes: ACMDM-S, ACMDM-B, ACMDM-L, and ACMDM-XL, corresponding to configurations with 8, 12, 16, and 20 layers and attention heads, respectively. This consistent scaling scheme enables systematic exploration of ACMDM's capacity and its effect on generation quality. In addition, we also vary the patch sizes for tokenization. We name different model variants according to their model and patch size (for tokenization); *e.g.*, ACMDM-XL-PS2 refers to the XL variant with a patch size of $1 \times 2$.

## 3.2 Adding Controls to Absolute Joint Coordinates Generation

Most prior methods face significant challenges in controllable motion generation due to their reliance on local-relative representations, which naturally misalign with user-provided absolute coordinates control signals. In contrast, our absolute coordinates representation removes this misalignment, enabling seamless integration of control without classifier guidance [16] and input optimization [41].

To enable controllable text-driven motion generation, such as trajectory conditioning and temporal/spatial editing with absolute joint coordinates, we follow prior works [102, 15, 14] and integrate a ControlNet [118]-style module into the ACMDM architecture. As shown in Figure 2 (b), the noised absolute coordinate latent is first tokenized via a 2D convolutional layer and then fed into both the main ACMDM and a parallel ControlNet module. At the same time, textual and timestep conditions are encoded and provided to both ACMDM and the ControlNet as conditioning embeddings. Separately, structured control signals (*e.g.*, joint trajectories or partial-body constraints) are processed through a dedicated ControlNet condition encoder. The ControlNet receives both the tokenized noised inputs as well as control-specific features in additive combination with the textual and timestep embeddings. These fused features generate residuals, which are injected into the main ACMDM backbone at each layers via additive fusion. This modulation enables the model to follow both semantic instructions and structural constraints. In addition to the standard $L_2$ reconstruction loss on the diffusion target, we also apply an $L_2$ loss between the model's prediction and the control signal. We also freeze the parameters of the main ACMDM and only train the ControlNet branch, which is initialized as copies of the main ACMDM blocks, similar to prior works [118, 102, 15, 14].

### 3.3 Generating Meshes with Absolute Coordinates Representation

Towards achieving vivid, animatable human avatars, joint representations are insufficient; when translated to meshes through fitting models, they often result in shaky body parts, unnatural hand motions, and missing flesh dynamics [91, 12, 15, 14]. Direct motion generation at the mesh level, however, largely falls behind joint counterparts, mainly due to the complexity of modeling mesh representations. Here, we show that our absolute, non-kinematic representation naturally extends to mesh vertices, which is seamlessly supported by ACMDM without major architectural changes.

In specific, we explore direct motion generation of SMPL-H [59] mesh vertices, where each frame is represented as a set of absolute 3D vertex coordinates with shape $(L, N_v, 3)$, where $N_v = 6890$ denotes the number of vertices. Unlike absolute joint coordinates, where the number of joints $N_j$ is typically small, directly training diffusion models on full-resolution mesh data with $N_v = 6890$ is computationally prohibitive and unstable. To address this, we incorporate a 2D mesh autoencoder based on the Fully Convolutional Mesh Autoencoder [130]. The encoder spatially compresses the input mesh sequence $(L, N_v, 3)$ into a latent representation of shape $(L, n_v, d_v)$, where we set $n_v = 28$ for diffusion modeling efficiency and reconstruction quality. Once mesh vertices are encoded, we reuse the ACMDM framework to perform motion diffusion in this latent mesh space. The resulting sequence is tokenized using patch sizes of $1 \times 28$ and processed with the same formulation as our joint-based ACMDM. In the experiment section, we show the flexibility and scalability of our approach for high-fidelity motion generation over mesh vertices as well in addition to human joints.

## 4 Experiment

### 4.1 Datasets, Training Setups, and Evaluation Protocols

**Datasets.** To fairly evaluate different ACMDM designs and compare against prior models, we adopt the widely used HumanML3D [25] benchmark for standard text-to-motion generation, downstream tasks such as text-driven trajectory-controlled generation and upper-body editing, and direct text-to-SMPL-H mesh motion generation. We also include text-to-motion evaluations on KIT-ML [78], reported in the Appendix. HumanML3D contains 14,616 motion sequences sourced from AMASS [65] and HumanAct12 [27], each paired with three textual descriptions (44,970 annotations in total). All motions are standardized to 20 FPS and capped at 10 seconds. It is augmented via mirroring and split into training, validation, and test sets using a standard 80%/15%/5% split.

**Training Setups.** All ACMDM variants are trained using the AdamW optimizer with $\beta_1 = 0.9$ and $\beta_2 = 0.99$. We use a batch size of 64 with a maximum sequence length of 196 frames. The learning rate is initialized at $2 \times 10^{-4}$ and linearly warmed up over the first 2,000 steps. We apply a learning rate decay by a factor of 0.1 at 50,000 iterations during the training of 500 epochs. We also use an exponential moving average (EMA) of model weights to improve training stability and performance. During inference, we apply classifier-free guidance (CFG) [30] $= 3$ for text-to-motion generation and upper-body editing, 2.5 for trajectory control, and 4.5 for text-to-SMPL-H mesh motion generation.

Table 1: **Ablation study of the design choices of ACMDM on the HumanML3D dataset.** The results indicate that kinematic-aware redundancy is not necessary. Instead, absolute coordinates motion representation can achieve high-quality motion generation with AdaLN conditioning, the velocity diffusion objective ($\mathbf{v}$), and latent space modeling.

| Motion Representation | Conditioning Mechanism | Motion AE | Diffusion Objective | FID↓ | R-Precision Top 1↑ | R-Precision Top 2↑ | R-Precision Top 3↑ | Matching↓ |
|---|---|---|---|---|---|---|---|---|
| Absolute+Redundancy | Concat | ✗ | $\mathbf{x}_0$ | $0.771^{\pm.020}$ | $0.441^{\pm.002}$ | $0.633^{\pm.003}$ | $\mathbf{0.738^{\pm.002}}$ | $3.632^{\pm.009}$ |
| | | | $\epsilon$ | $0.868^{\pm.030}$ | $0.358^{\pm.003}$ | $0.538^{\pm.005}$ | $0.650^{\pm.004}$ | $4.168^{\pm.025}$ |
| | | | $\mathbf{v}$ | $\mathbf{0.276^{\pm.006}}$ | $\mathbf{0.445^{\pm.002}}$ | $\mathbf{0.634^{\pm.002}}$ | $\mathbf{0.738^{\pm.002}}$ | $\mathbf{3.613^{\pm.008}}$ |
| Absolute | Concat | ✗ | $\mathbf{x}_0$ | $0.969^{\pm.029}$ | $0.356^{\pm.003}$ | $0.539^{\pm.004}$ | $0.648^{\pm.003}$ | $4.362^{\pm.013}$ |
| | | | $\epsilon$ | $0.419^{\pm.013}$ | $0.436^{\pm.002}$ | $0.630^{\pm.003}$ | $0.736^{\pm.003}$ | $3.717^{\pm.013}$ |
| | | | $\mathbf{v}$ | $\mathbf{0.208^{\pm.012}}$ | $\mathbf{0.451^{\pm.003}}$ | $\mathbf{0.643^{\pm.003}}$ | $\mathbf{0.751^{\pm.002}}$ | $\mathbf{3.544^{\pm.010}}$ |
| Absolute | AdaLN | ✗ | $\mathbf{x}_0$ | $0.133^{\pm.004}$ | $0.485^{\pm.002}$ | $0.680^{\pm.002}$ | $0.779^{\pm.002}$ | $3.386^{\pm.012}$ |
| | | | $\epsilon$ | $0.125^{\pm.007}$ | $0.493^{\pm.003}$ | $0.685^{\pm.003}$ | $0.783^{\pm.002}$ | $3.343^{\pm.009}$ |
| | | | $\mathbf{v}$ | $\mathbf{0.121^{\pm.006}}$ | $\mathbf{0.502^{\pm.002}}$ | $\mathbf{0.692^{\pm.003}}$ | $\mathbf{0.789^{\pm.003}}$ | $\mathbf{3.304^{\pm.008}}$ |
| Absolute | AdaLN | Causal AE | $\mathbf{x}_0$ | $0.137^{\pm.007}$ | $0.473^{\pm.002}$ | $0.670^{\pm.002}$ | $0.772^{\pm.003}$ | $3.451^{\pm.011}$ |
| | | | $\epsilon$ | $0.188^{\pm.006}$ | $0.475^{\pm.003}$ | $0.670^{\pm.002}$ | $0.775^{\pm.002}$ | $3.393^{\pm.012}$ |
| | | | $\mathbf{v}$ | $\mathbf{0.109^{\pm.005}}$ | $\mathbf{0.508^{\pm.002}}$ | $\mathbf{0.701^{\pm.003}}$ | $\mathbf{0.798^{\pm.003}}$ | $\mathbf{3.253^{\pm.010}}$ |
| Absolute | AdaLN | Non-Causal VAE | $\mathbf{v}$ | $0.178^{\pm.006}$ | $0.497^{\pm.002}$ | $0.687^{\pm.003}$ | $0.785^{\pm.004}$ | $3.323^{\pm.010}$ |
| | | Non-Causal AE | $\mathbf{v}$ | $0.150^{\pm.005}$ | $0.502^{\pm.003}$ | $0.693^{\pm.003}$ | $0.787^{\pm.003}$ | $3.296^{\pm.010}$ |
| | | Causal VAE | $\mathbf{v}$ | $0.115^{\pm.005}$ | $0.504^{\pm.002}$ | $0.697^{\pm.002}$ | $0.795^{\pm.003}$ | $3.278^{\pm.011}$ |

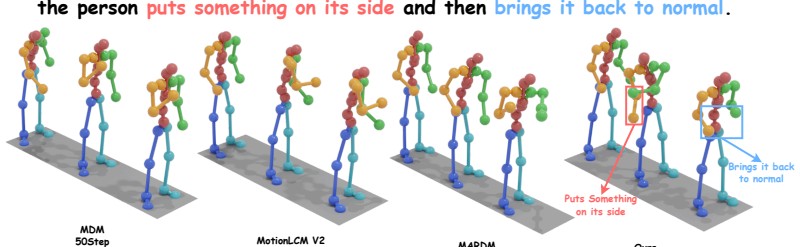

the person puts something on its side and then brings it back to normal.

MDM 50Step  MotionLCM V2  MARDM  Ours

Figure 3: **Visual comparisons of generated motion between ACMDM and state-of-the-art methods.** ACMDM generates more realistic motion that accurately follows the textual condition.

**Evaluation Metrics.** We adopt the robust evaluation framework proposed by [66], focusing on essential, animatable motion features. Following [25, 66], we report: (1) R-Precision (Top-1/2/3) and Matching (semantic alignment with captions); (2) FID (distribution similarity); (3) MultiModality (motion diversity per prompt); and (4) CLIP-Score (cosine similarity between motion and caption embeddings). For trajectory-control evaluations [42], we additionally report Diversity (variability within generated motions), Foot Skating Ratio, Trajectory Error, Location Error, and Average Joint Error (accuracy of controlled joints at keyframes). Metrics are averaged over five levels of control intensity (1%, 2%, 5%, 25%, 100%). During training, control intensity levels are randomly sampled. For direct SMPL-H mesh generation, we also report Laplacian Surface Distance (LSD) to assess mesh structural preservation relative to the ground-truth T-pose. More metric details are in Appendix.

## 4.2 Ablating ACMDM Designs

**Necessity of Kinematic-aware and Redundant Motion Representation.** Prior attempts [102] of text-to-absolute-coordinate motion generation adopt InterGen [52]'s representation with heavy kinematic-aware redundancy and the $\mathbf{x}_0$ objective, but result in unrealistic motion. To systematically analyze this, in the top two sections of Table 1, we train an ACMDM-S-PS22 variant. We match the model size and flattened spatial embedding style used in prior works in two settings: one using absolute coordinates with kinematic-aware and redundant representation (*i.e.*, InterGen's representation), and another using plain absolute coordinates (our proposed). The results show that while the previously widely adopted $\mathbf{x}_0$-prediction diffusion benefits slightly from the redundancy, velocity prediction ($\mathbf{v}$) with plain absolute coordinates (our proposed) achieves better performance. Notably, by modeling plain absolute coordinates with $\mathbf{v}$ prediction, ACMDM achieves a FID that is **0.563 lower** and an R-Precision Top-3 score that is **0.013 higher** compared to redundant $\mathbf{x}_0$ prediction. These results demonstrate that with a more suitable diffusion objective ($\mathbf{v}$ prediction), and the previously assumed necessary kinematic-aware redundancy is not required for achieving high-quality motion generation. Therefore, for the rest of the paper, all ACMDM models will adopt the pure absolute coordinates representation without any kinematic-aware or redundant features.

Table 2: **Quantitative text-to-motion evaluation.** We repeat the evaluation 20 times and report the average with 95% confidence interval. We use **bold** face / underline to indicate the best/2$^{nd}$ results.

| Methods | FID↓ | R-Precision↑ | | | Matching↓ | MModality↑ | CLIP-score↑ |
|---|---|---|---|---|---|---|---|
| | | Top 1 | Top 2 | Top 3 | | | |
| **Real** | $0.000^{\pm.000}$ | $0503^{\pm.002}$ | $0.696^{\pm.001}$ | $0.795^{\pm.002}$ | $3.244^{\pm.005}$ | - | $0.639^{\pm.001}$ |
| MDM-50Step [91] | $0.518^{\pm.032}$ | $0.440^{\pm.007}$ | $0.636^{\pm.006}$ | $0.742^{\pm.004}$ | $3.640^{\pm.028}$ | $\mathbf{3.604}^{\pm.031}$ | $0.578^{\pm.003}$ |
| MotionDiffuse [119] | $0.778^{\pm.005}$ | $0.450^{\pm.006}$ | $0.641^{\pm.005}$ | $0.753^{\pm.005}$ | $3.490^{\pm.023}$ | $\underline{3.179}^{\pm.046}$ | $0.606^{\pm.004}$ |
| ReMoDiffuse [120] | $0.883^{\pm.021}$ | $0.468^{\pm.003}$ | $0.653^{\pm.003}$ | $0.754^{\pm.005}$ | $3.414^{\pm.020}$ | $2.703^{\pm.154}$ | $0.621^{\pm.003}$ |
| MLD++ [14] | $2.027^{\pm.021}$ | $0.500^{\pm.003}$ | $0.691^{\pm.002}$ | $0.789^{\pm.001}$ | $3.220^{\pm.008}$ | $1.924^{\pm.065}$ | $0.639^{\pm.002}$ |
| MotionLCM V2 [14] | $2.267^{\pm.023}$ | $0.501^{\pm.002}$ | $0.693^{\pm.002}$ | $0.790^{\pm.002}$ | $3.192^{\pm.009}$ | $1.780^{\pm.062}$ | $0.640^{\pm.003}$ |
| MARDM [66]-$\epsilon$ | $0.116^{\pm.004}$ | $0.492^{\pm.006}$ | $0.690^{\pm.005}$ | $0.790^{\pm.005}$ | $3.349^{\pm.010}$ | $2.470^{\pm.053}$ | $0.637^{\pm.005}$ |
| MARDM [66]-$\mathbf{v}$ | $0.114^{\pm.007}$ | $0.500^{\pm.004}$ | $0.695^{\pm.003}$ | $0.795^{\pm.003}$ | $3.270^{\pm.009}$ | $2.231^{\pm.071}$ | $0.642^{\pm.002}$ |
| **ACMDM-S-PS22** | $\underline{0.109}^{\pm.005}$ | $\underline{0.508}^{\pm.002}$ | $\underline{0.701}^{\pm.003}$ | $\underline{0.798}^{\pm.003}$ | $3.253^{\pm.010}$ | $2.156^{\pm.061}$ | $\underline{0.642}^{\pm.001}$ |
| **ACMDM-XL-PS2** | $\mathbf{0.058}^{\pm.004}$ | $\mathbf{0.522}^{\pm.002}$ | $\mathbf{0.713}^{\pm.002}$ | $\mathbf{0.807}^{\pm.002}$ | $\mathbf{3.205}^{\pm.008}$ | $2.077^{\pm.083}$ | $\mathbf{0.652}^{\pm.001}$ |

**Concatenation *vs.* AdaLN.** In the third section of Table 1, we switch from the widely adopted concatenation-based conditioning to AdaLN conditioning with an ACMDM-S-PS22 variant with pure absolute coordinates. Our results show that across all diffusion objectives, better conditioning mechanism (AdaLN) lead to significant improvements. Notably, with **v** prediction, ACMDM achieves an FID of **0.121** and an R-Precision Top-3 score of **0.789**, substantially outperforming concatenation-based conditioning. These findings demonstrate that an effective conditioning mechanism is a key factor in achieving high-quality motion generation. Therefore, for all subsequent experiments, we adopt AdaLN-based conditioning mechanism across all ACMDM models.

**Raw Absolute Coordinates *vs.* Latent Space.** In the fourth section of Table 1, we switch from directly modeling raw absolute coordinates to a latent space. Our results show that latent space modeling further improves generation quality while also offering faster inference for **v** prediction, achieving the best FID of **0.109** and R-Precision Top-3 score of **0.798**

We additionally compare different AutoEncoder variants: Causal-AE, Non-Causal-AE, and VAE in the last section of Table 1. Among them, Causal-AE achieves the best overall performance. Therefore, for all subsequent experiments, we adopt Causal-AE as our default setup. Since velocity (**v**) prediction consistently yields the best performance across all settings, we also adopt it as the default diffusion objective.

**Scaling Model and Decreasing Patch Sizes.** In Figure 4, we train 12 ACMDM models over all model configs (S, B, L, XL) and patch sizes $(1 \times 22, 1 \times 11, 1 \times 2)$. In all cases, we find that increasing model size and decreasing patch size lead to improved text-to-motion generation performance both with and without CFG across all metrics. Notably, ACMDM-XL-PS2 achieves an FID of **0.058** and an R-Precision Top-1 score of **0.522**, outperforming the most recent state-of-the-art MARDM by **0.056** in FID and **0.022**

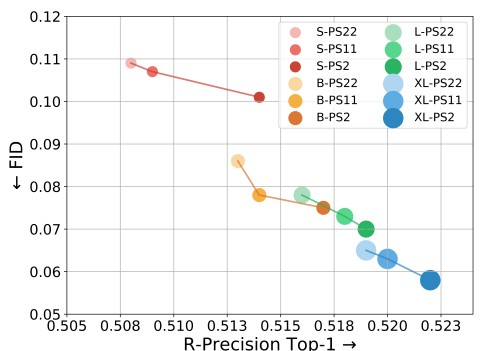

Figure 4: **Scaling of ACMDM with model capacity and decreasing patch size.** We use red for S, orange for B, green for L, and blue for XL, with color gradients indicating decreasing patch sizes. ACMDM exhibits strong scalability, with performance consistently improving as model size increases and patch size decreases.

in R-Precision Top-1. These findings demonstrate the effectiveness of scaling model capacity and decreasing patch sizes with absolute joint coordinates. We include detailed results in Appendix.

### 4.3 Comparison to State-of-the-Art Text-to-Motion Generation Methods

We present the quantitative comparison between our method and state-of-the-art text-to-motion generation baselines in Table 2, as well as qualitative comparison in Figure 3 and Appendix. As observed, our method achieves superior performance across multiple key metrics, including FID, R-Precision, Matching Score, and CLIP-Score. Compared to existing approaches, ACMDM demonstrates a significantly stronger ability to generate high-fidelity, semantically aligned motions that closely follow textual instructions. Notably, even for our smallest ACMDM variant, ACMDM-S-PS22, it outperforms all prior state-of-the-art methods. Larger ACMDM models, such as ACMDM-XL-PS2, further amplify the performance gains across all evaluation metrics.

Table 3: **Quantitative text-conditioned motion generation with spatial control signals and upper-body editing on HumanML3D.** In the first section, methods are trained and evaluated solely on pelvis controls. In the middle section, methods are trained on all joints and evaluated separately on each controlled joint. Only average results are reported for brevity. We include details in Appendix. Last section presents upper-body editing results. **bold** face / underline indicates the best/2$^{nd}$ results.

| Controlling Joint | Methods | AITS↓ | Classifier Guidance | FID↓ | R-Precision Top 3 | Diversity→ | Foot Skating Ratio↓ | Traj. err.↓ | Loc. err.↓ | Avg. err.↓ |
|---|---|---|---|---|---|---|---|---|---|---|
| | GT | − | - | 0.000 | 0.795 | 10.455 | - | 0.000 | 0.000 | 0.000 |
| **Train On Pelvis** | MDM [91] | 16.34 | ✗ | 1.792 | 0.673 | 9.131 | 0.1019 | 0.4022 | 0.3076 | 0.5959 |
| | PriorMDM [83] | 20.19 | ✗ | 0.393 | 0.707 | 9.847 | 0.0897 | 0.3457 | 0.2132 | 0.4417 |
| | GMD [42] | 137.63 | ✓ | 0.238 | 0.763 | 10.011 | 0.1009 | 0.0931 | 0.0321 | 0.1439 |
| | OmniContol [102] | 81.00 | ✓ | 0.081 | 0.789 | 10.323 | **0.0547** | 0.0387 | 0.0096 | 0.0338 |
| | MotionLCM V2+CtrlNet [14] | **0.066** | ✗ | 3.978 | 0.738 | 9.249 | 0.0901 | 0.1080 | 0.0581 | 0.1386 |
| | **ACMDM-S-PS22+CtrlNet** | 2.51 | ✗ | 0.067 | **0.805** | **10.481** | 0.0591 | **0.0075** | **0.0010** | **0.0100** |
| **Train On All Joints (Average)** | OmniContol [102] | 81.00 | ✓ | 0.126 | 0.792 | 10.276 | 0.0608 | 0.0617 | 0.0107 | 0.0404 |
| | MotionLCM V2+CtrlNet [14] | **0.066** | ✗ | 4.504 | 0.715 | 9.230 | 0.1119 | 0.2740 | 0.1315 | 0.2464 |
| | **ACMDM-S-PS22+CtrlNet** | 2.51 | ✗ | **0.070** | **0.803** | **10.526** | **0.0596** | **0.0117** | **0.0019** | **0.0197** |

| Controlling Joint | Methods | AITS↓ | Classifier Guidance | FID↓ | R-Precision Top 1 | R-Precision Top 2 | R-Precision Top 3 | Matching↓ | Diversity→ | - |
|---|---|---|---|---|---|---|---|---|---|---|
| **UpperBody Edit** | MDM [91] | 16.34 | ✗ | 1.918 | 0.359 | 0.556 | 0.654 | 4.793 | 9.210 | |
| | OmniControl [119] | 81.00 | ✓ | 0.909 | 0.428 | 0.614 | 0.722 | 3.694 | 10.207 | |
| | MotionLCM V2+CtrlNet [119] | **0.066** | ✗ | 3.922 | 0.404 | 0.592 | 0.692 | 5.610 | 9.309 | |
| | **ACMDM-S-PS22+CtrlNet** | 2.51 | ✗ | 0.076 | 0.532 | 0.719 | 0.820 | 3.098 | 10.586 | |

Table 4: **Quantitative results** for direct text-to-SMPL-H mesh motion generation on HumanML3D.

| Size | Transformer | FID ↓ | R-Precision Top 1 ↑ | R-Precision Top 2 ↑ | R-Precision Top 3 ↑ | Matching↓ | CLIP-score↑ | LSD↓ |
|---|---|---|---|---|---|---|---|---|
| S | 8 head 512 dim | $0.211^{\pm.005}$ | $0.478^{\pm.004}$ | $0.682^{\pm.003}$ | $0.784^{\pm.003}$ | $3.405^{\pm.011}$ | $0.620^{\pm.002}$ | $0.0026^{\pm.0002}$ |
| B | 12 head 768 dim | $0.181^{\pm.003}$ | $0.490^{\pm.003}$ | $0.691^{\pm.003}$ | $0.783^{\pm.002}$ | $3.345^{\pm.010}$ | $0.631^{\pm.001}$ | $\mathbf{0.0024^{\pm.0002}}$ |
| L | 16 head 1024 dim | $0.160^{\pm.004}$ | $0.497^{\pm.003}$ | $0.696^{\pm.002}$ | $0.790^{\pm.002}$ | $3.341^{\pm.009}$ | $0.633^{\pm.0}$ | $0.0025^{\pm.0001}$ |
| XL | 20 head 1280 dim | $\mathbf{0.139^{\pm.003}}$ | $\mathbf{0.498^{\pm.003}}$ | $\mathbf{0.704^{\pm.003}}$ | $\mathbf{0.794^{\pm.003}}$ | $\mathbf{3.309^{\pm.007}}$ | $\mathbf{0.636^{\pm.001}}$ | $0.0025^{\pm.0001}$ |

## 4.4 Comparison to State-of-the-Art Controllable Motion Generation Methods

We present quantitative comparisons between our method and state-of-the-art methods on text-driven trajectory control and upper-body editing in Table 3. For the trajectory control task, prior works [42, 102, 14] have shown that inference-time classifier guidance is crucial for achieving strong control performance. However, we show that even with our smallest ACMDM variant that matches to baseline model sizes and embedding formats, our absolute coordinate formulation achieves superior motion fidelity and control accuracy without the need for time-consuming classifier guidance from control signals. This results in significantly faster generation compared to guidance-dependent approaches (**2.51** *v.s.* **81.0** seconds). For the upper-body editing task, we follow the evaluation protocol proposed by [77, 75], where we fix the pelvis, left foot, and right foot joints and edit the upper body motion according to textual prompts. Our method achieves substantially better generation quality across all evaluation metrics, validating the effectiveness of our proposed approach.

## 4.5 Evaluations on Absolute Mesh Vertex Coordinates Motion Generation

We evaluate ACMDM on SMPL-H absolute mesh vertex coordinates motion generation in Table 4. We train and compare four ACMDM model sizes—S, B, L, and XL, with the patch size of $1 \times 28$. Despite the significantly increased complexity of modeling full mesh sequences compared to joint sequences, our ACMDM models still achieve strong performance. Notably, all variants achieve results competitive with the best text-to-joint generation models, while operating directly on high-dimensional vertex spaces. This highlights the effectiveness and flexibility of our absolute coordinates motion representation in handling broader motion generation tasks beyond human joints.

## 5 Conclusion

In conclusion, we presented ACMDM, a novel text-driven motion diffusion framework built on an absolute coordinates motion representation. We run extensive analysis to identify an optimal setting, including the velocity prediction diffusion objective, optimized conditioning mechanisms (AdaLN), and latent motion representation. Our model naturally supports downstream control tasks, which removes the misalignment between local motion representation and absolute controlling, and also generalizes to direct SMPL-H mesh vertices motion generation. Extensive experiments demonstrate that ACMDM achieves superior performance and scalability across text-to-motion benchmarks.

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
