# OpenReview forum: "Absolute Coordinates Make Motion Generation Easy"
_NeurIPS.cc/2025/Conference — Submitted to NeurIPS 2025_

### Official Review · Reviewer_oZAK · 2025-06-29

**Clarity:** 2
**Significance:** 2
**Originality:** 2
**Rating:** 3
**Confidence:** 4

**Summary:**

This paper challenges the prevailing paradigm in text-to-motion generation, which heavily relies on the local-relative, kinematic-aware motion representations popularized by the HumanML3D dataset. The authors argue that this standard representation, while beneficial for earlier models, imposes critical limitations on modern diffusion-based approaches and complicates downstream applications. The central contribution is a return to a "radically simplified and long-abandoned" alternative: representing human motion using absolute joint coordinates in a global 3D space.
The proposed method, the Absolute Coordinates Motion Diffusion Model (ACMDM), is built upon a simple Transformer backbone. The paper's focus is not on architectural novelty but on a systematic investigation of the design choices that enable this simplified representation to succeed. Through extensive ablation studies, the authors demonstrate that a specific combination of techniques is key: using absolute coordinates with AdaLN conditioning for text and timestep embeddings, and adopting a velocity (v) prediction objective for the diffusion process. The use of a motion autoencoder to operate in a latent space is also shown to further improve performance.
The authors claim three primary contributions. First, that this simplified framework achieves state-of-the-art performance on standard text-to-motion benchmarks (HumanML3D and KIT-ML) without requiring complex architectural components or auxiliary kinematic losses. Second, that the absolute representation inherently simplifies and improves performance on downstream tasks like motion control and editing, removing the need for task-specific engineering and computationally expensive classifier guidance common in prior work. Third, the paper demonstrates the framework's promising generalization to the direct generation of high-dimensional SMPL-H mesh vertex coordinates, a challenging task that has been largely unexplored in the literature.

**Questions:**

Please see weaknesses sections. The authors could try to answer to point 1,2,4,5, 6 but I don’t think I will change my final rating. In my opinion though the idea may have merit the paper’s presentation especially the video and figures are sub-par.

**Ethical Concerns:**

["NO or VERY MINOR ethics concerns only"]

**Final Justification:**

To summarize my opinion is that 1) the paper has some interesting insights but 2) the presentations is below NeurIPS standards -a) there are not enough figures b) there should be one unified video - not just an html file 3) the mesh based visualization which is claimed to be a benefit of the method should have been used for a majority of the visualizations which in the current html file is not 4) there are missing references which do not give enough credit to existing work. As such I will give a rating leaning towards rejection - though I won't be arguing very strongly for rejection. If the other reviewers are strongly for it going in, I am not going to argue strongly against it.

**Limitations:**

yes to a certain extent.

**Quality:**

1

**Strengths And Weaknesses:**

Strengths:

1. The paper’s primary strength is its fundamental re-evaluation of a core assumption that has dominated the text-to-motion field for years. The local-relative, kinematic-aware representation introduced with HumanML3D has become the de facto standard, adopted by nearly all recent state-of-the-art models. This paper boldly argues that this convention is not only unnecessary but detrimental for modern architectures..
2. The ablations are well done and most of the design choices are evaluated with rigour.
3. The paper makes a powerful case for the practical significance of its approach, particularly for controllable motion generation. By representing motion in absolute coordinates, the model's output space is inherently aligned with the absolute spatial control signals (e.g., trajectories, key-pose constraints) that users typically provide. This stands in stark contrast to prior state-of-the-art methods like OmniControl, which are built on relative representations and must resort to complex and computationally expensive techniques like classifier guidance to enforce absolute spatial constraints.
4. The demonstration of directly generating the motion of SMPL-H mesh vertices (N) without time consuming IK steps as required in current pipelines

WEAKNESS
1. Strange Video Format: There is not one video in the supp material but the video is submitted in pieces - with separate folders for different results. This is very unusual. With this kind of presentation, it becomes difficult for me to recommend acceptance at a conference of the caliber of NeurIPS.
2. Video Results:For motion control, results are only shown on joints - no meshes - comparison with omnicontrol only includes joints, which is very surprising because one of the claims of the paper is that it allows for direct prediction of SMPLH vertices so why use joints

3. Insufficient visualizations - there are a acouple of quantitative visualizations in the main paper. This is a 3D paper and again I don’t see how one can accept a 3D paper with such limited quantitative results.

4. Missing citations: There is some merit in the idea of using absolute coordinates. I agree with the authors that the representation used in MDM and follow ups is redundant and could be refined. But the rep proposed in the paper has been used before - hang et al. MOJO - CVPR 21, Zhao et al. DartControl ICLR 25. Both these papers are not cited. There is still novelty in the current submission as its been used in the context of motion diffusion models but there is a major weakness and again because of this I cannot recommend acceptance. The authors need to properly discuss how their works differs from MOJO DartControl - and why their paper differs signifcantly

5. Questionable Choice of Evaluation Metrics: This weakness is directly related to the first and is equally critical. The authors state, "We adopt the robust evaluation framework proposed by ". This is a misleading statement. The MARDM paper explicitly argues that the standard evaluation framework is  not robust and, in fact, proposes a new, "more robust evaluation method" to rectify its flaws. The critique presented in MARDM is that standard metrics like FID and R-Precision are calculated over all motion features, including redundant ones. This setup, they demonstrate, unfairly penalizes continuous diffusion models (which struggle to perfectly model the noise in these non-essential dimensions) while favoring VQ-based discrete models (whose codebooks enforce consistency across all dimensions). The authors of MARDM show that the performance gap between these model classes narrows significantly when evaluation is restricted to only the essential, animatable features. By choosing to use the older, flawed metrics instead of the "more robust" ones proposed in the very paper they cite for their protocol, the authors of the present work undermine the credibility of their state-of-the-art claims. This choice requires strong justification, as it could be perceived as avoiding a more rigorous and fair evaluation.

6. Insufficient Analysis of Kinematic Plausibility and Failure Modes: The paper correctly notes that early attempts to use absolute coordinates failed due to the generation of "unnatural, non-human-like motions". The historical dominance of relative representations stems from their ability to implicitly enforce kinematic constraints like constant bone lengths and natural joint limits. This paper claims to overcome this challenge without any explicit kinematic losses, relying on the model to learn these constraints implicitly from data. While the low FID scores suggest that the  distribution of generated motions is close to the real data, this does not guarantee that every individual sample is kinematically plausible. The paper lacks a dedicated analysis of potential failure modes. For instance, does the model ever produce motions with stretched limbs, disjointed body parts, or other physically impossible artifacts, especially when faced with unusual or out-of-distribution text prompts? A qualitative analysis of failure cases, or a quantitative metric tracking bone-length variance, would be necessary to fully substantiate the claim that this fundamental weakness of absolute coordinates has been resolved. This is particularly critical for the mesh generation results, where small joint-level errors can be magnified into significant visual artifacts on the mesh surface.

---

> ### Author Rebuttal · Authors · 2025-07-30
>
> # **To Reviewer oZAK:**
> &nbsp;&nbsp;&nbsp;&nbsp;&nbsp;&nbsp;Thank you for your detailed feedback and your recognition of our main motivation, ablation study designs, and the practical benefits of absolute coordinate representation for motion control and direct SMPL-H mesh generation. Below, we provide detailed responses to the weaknesses and questions you raised.
>
> ## **Q1: Regarding Strange Video Format.**
> &nbsp;&nbsp;&nbsp;&nbsp;&nbsp;&nbsp;We thank the reviewer for the feedback and the concern regarding the video format. We would like to clarify that our qualitative results are presented through a locally browsable HTML interface, as mentioned in the Appendix Section J. In this way, readers can simply double-click the index.html file for full visualizations. The separate data folders are used solely to organize the supporting files for this interface. This design enables readers to interactively browse extensive results by task (e.g., text-to-motion, motion control, direct mesh generation) without the need to sit through a long video. We chose this format to allow faster navigation, side-by-side comparisons, and broader qualitative coverage, which are often not feasible with traditional one linear video.
>
> ## **Q2: Response to Joint *v.s.* Mesh Video Results for Motion Control Comparisons.**
> &nbsp;&nbsp;&nbsp;&nbsp;&nbsp;&nbsp;We thank the reviewer for the observation. The reason we visualize joint-level visualization (rather than full meshes) in our motion control results is because we follow the experimental setup defined by prior work OmniControl \[102\], which specifically formulates the control task at the joint level (targeting key joints such as the pelvis, feet, head, and wrists). All baseline methods are not designed to operate on mesh-level control, and we adhere to this protocol to ensure a fair and consistent comparison.
> &nbsp;&nbsp;&nbsp;&nbsp;&nbsp;&nbsp;Additionally, we intentionally use joint-space visualizations to better showcase the fine-grained controllability of our model. Mesh-based visualizations can obscure the accuracy of control due to the surrounding body volume around the target joint, making it harder to judge whether the control targets are precisely achieved. In contrast, joint-level visualization offers a clearer and more accurate visualization of whether the model successfully reaches the intended control targets, especially when comparing across methods.
>
> ## **Q3: Response to Insufficient Visualization.**
> &nbsp;&nbsp;&nbsp;&nbsp;&nbsp;&nbsp;We thank the reviewer for the comment. We would like to clarify that motion data is inherently 4D where each frame contains 3D spatial coordinates, and the temporal dimension is equally critical to understanding motion quality. Static figures in the paper cannot effectively convey this temporal structure, especially for evaluating smoothness, realism, and controllability.
> &nbsp;&nbsp;&nbsp;&nbsp;&nbsp;&nbsp;Nevertheless, we include two key visualizations in the main paper (a teaser and a side-by-side comparison). To complement this, we also provide an extensive set of results in the supplementary HTML webpage, covering over 45 motion clips across three distinct tasks. This choice reflects our belief that video results are essential for evaluating the temporal dynamics and overall quality of generated motion. We hope the reviewer will consider the full scope of visualizations included in the video results as part of the overall evaluation.
>
> ## **Q4: Regarding MOJO, DartControl, and Their Representations.**
> &nbsp;&nbsp;&nbsp;&nbsp;&nbsp;&nbsp;We thank the reviewer for highlighting these important prior works. We will include MOJO in the revised version of the paper. However, DARTControl is in fact cited as \[125\] in our current submission.
> &nbsp;&nbsp;&nbsp;&nbsp;&nbsp;&nbsp;We would like to clarify how our approach differs from both works. First, MOJO (CVPR 2021\) predates HumanML3D \[25\] (CVPR 2022), and primarily focuses on a different task: human motion prediction. They adopt surface keypoints as their representation, rather than joint coordinates or full mesh vertices. This also aligns with our statement in the introduction section that the field largely converged on root-relative joint representations following the introduction of HumanML3D. Our framing of absolute coordinates as “simple and long-abandoned” reflects this trajectory. We do not claim the representation is novel, but rather that it has been overlooked for text-to-motion in recent years and deserves rethinking and empirical re-examination. Second, DARTControl uses a relative representation in the form of SMPL-X \[59\] parameters, which differ fundamentally from our formulation of absolute coordinates.
>
> ## **Q5: Response to Questionable Choice of Evaluation Metrics.**
> &nbsp;&nbsp;&nbsp;&nbsp;&nbsp;&nbsp;We fully agree with the reviewer that the evaluation framework proposed in MARDM \[66\] provides a more robust and fair assessment by focusing on essential, animatable dimensions and excluding redundant features. We would like to point out that our results were, in fact, obtained using the same evaluation method and protocol as MARDM, and results exactly following MARDM.
>
> ## **Q6: Regarding Insufficient Analysis of Kinematic Plausibility and Failure Modes.**
> &nbsp;&nbsp;&nbsp;&nbsp;&nbsp;&nbsp;We argue that since FID compares the distribution of generated motions against real human motion, samples with highly unnatural or kinematically impossible artifacts would significantly increase the distance between these distributions. Therefore, the consistently low FID scores achieved by our model suggest that such failure cases are rare.
> &nbsp;&nbsp;&nbsp;&nbsp;&nbsp;&nbsp;We would also like to clarify a common misconception: relative joint positions do not inherently guarantee constant bone lengths. It is possible to predict arbitrarily small or large relative offsets that violate bone length constraints. Only rotational representations (e.g., joint rotations) inherently preserve bone lengths with a fixed skeleton template. However, the rotation data in HumanML3D is known to be redundant and inaccurate (as discussed in our paper Line 28, and noted in HumanML3D’s GitHub issue). Consequently, no baseline method using HumanML3D’s relative joint positions can strictly preserve bone length.
> &nbsp;&nbsp;&nbsp;&nbsp;&nbsp;&nbsp;To evaluate ACMDM’s ability to maintain bone length consistency, we conducted two quantitative tests on the HumanML3D testset:
> &nbsp;&nbsp;&nbsp;&nbsp;&nbsp;&nbsp;1) Intra-sequence bone length stability – measuring changes in bone lengths within a generated sequence.
> &nbsp;&nbsp;&nbsp;&nbsp;&nbsp;&nbsp;(2) Bone length consistency with ground truth – comparing generated bone lengths against those from the ground truth.
> &nbsp;&nbsp;&nbsp;&nbsp;&nbsp;&nbsp;We also compared ACMDM to SOTA relative representation baselines with results presented in Table A4. Results indicate that ACMDM **not only maintains stable bone lengths but also outperforms some relative baselines** such as MotionLCM V2 \[14\] and MARDM \[14\].
> &nbsp;&nbsp;&nbsp;&nbsp;&nbsp;&nbsp;Due to the NeurIPS rebuttal policy, we cannot include additional videos, but we observed no noticeable bone length artifacts in the HumanML3D testset. To further probe robustness, we tested ACMDM on 10 out-of-distribution prompts involving “flying” (e.g., “a person is flying in the air”), which are impossible motions. Despite the unusual prompts, the generated motions mostly showed realistic behavior (e.g., standing, waving) rather than attempting unrealistic "flying" motion, and intra-sequence bone length variation remained very low (0.0071 m), confirming the reliability of our approach. We will provide extended qualitative results with failure case visualizations in the revised version.
>
> &nbsp;&nbsp;&nbsp;&nbsp;&nbsp;&nbsp;**Table A4: Bone length consistency on the humanml3d testset.**
> | Method | Intra-Seq Variation (m) $\\downarrow$ | *v.s.* GT Bone Length (m) $\\downarrow$ |
> | :---- | :---- | :---- |
> | MLD++ \[14\] | 0.0053 | 0.0057 |
> | MotionLCM V2 \[14\] | 0.0077 | 0.0073 |
> | MARDM-v \[66\] | 0.0070 | 0.0071 |
> | ACMDM-S-PS22 | 0.0061 | 0.0069 |
> | ACMDM-XL-PS2 | 0.0059 | 0.0065 |

---

> ### Comment · Reviewer_oZAK · 2025-08-05
> **response after rebuttal**
>
> I thank the authors for putting in the effort towards this rebuttal. Thanks for the pointer towards the html file. It did not work with safari on my laptop and hence my comments about video format. However I tried it again with chrome this time and it worked. The video files indeed look much better when visualized using the html file. It would make sense in the future to provide an mp4 file for the whole video- just to make sure these troubles dont happen. I acknowledge the authors responses 1,5,6 addresses the queries I raised. However I think 2,3,4 remain valid concerns. MOJO should be cited properly in the paper and acknowledgement given to papers that use pointcloud representation for motion synthesis. Regarding 3, I think that the paper falls slightly short of the presentation standards expected at Neurips. For 2, the authors are right to point out that a mesh obscures the fine grained control that can be observed with Joint Positions however the authors claim text-to-motion - not spatially controlled motion -  is their main task which does not require fine-grained details and I find it very surprising that more mesh visualizations are not provided as they claim that direct prediction of mesh vertices is a benefit of this paper. I am still inclined towards a rejection but I will downweight my vote from a strong reject to a boderline reject - the upgrade is primarily because the html file started working.

---

> > ### Author Response · Authors · 2025-08-05
> >
> > # Dear Reviewer oZAK:
> >
> > &nbsp;&nbsp;&nbsp;&nbsp;&nbsp;&nbsp;Thank you very much for your thoughtful feedback and for engaging further in the discussion. We are glad our responses helped clarify concerns 1, 5, and 6, and we sincerely appreciate your willingness to increase your rating.
> >
> > &nbsp;&nbsp;&nbsp;&nbsp;&nbsp;&nbsp;We would like to take this opportunity to clarify a few remaining points:
> >
> > &nbsp;&nbsp;&nbsp;&nbsp;&nbsp;&nbsp;On **Concern 2**, thank you for acknowledging our response regarding your original concern about the visualizations on the spatially controllable motion generation task. In response to your new comment, we would like to clarify that controllable motion generation is indeed one of our main contributions, alongside joint-level text-to-motion generation and direct SMPL-H mesh generation—all of which are supplemented with visualizations in the HTML file. For the joint-level text-to-motion task, we visualize joint-level skeletons in accordance with the skeletal task design defined in \[25\], as all baseline methods directly output joints rather than meshes. Skeleton-based visualization is also a common practice, as seen in prior works such as MotionDiffuse \[119\] and MARDM \[14\]. At the same time, we also provide mesh-level visualizations to demonstrate the potential of our method for direct mesh generation and to distinguish the differing nature of the two tasks. We will include additional mesh visualizations in the paper and on the supplementary website to further emphasize this capability.
> >
> > &nbsp;&nbsp;&nbsp;&nbsp;&nbsp;&nbsp;Regarding **Concern 3**, we agree that presentation standards are important. We included two key visualizations in the main paper and over 45 motion sequences in the supplementary HTML, as we believe the 4D nature and temporal dynamics of motion are best conveyed through video rather than static figures. We hope this design choice reflects our effort to present the work as clearly and comprehensively as possible.
> >
> > &nbsp;&nbsp;&nbsp;&nbsp;&nbsp;&nbsp;Finally, regarding MOJO citation in **Concern 4**, we appreciate you bringing up MOJO and will cite it in the revised version as promised in our rebuttal. As noted previously in our rebuttal, we would like to clarify that MOJO predates HumanML3D, utilizes surface keypoints, and addresses a different task—motion prediction rather than text-to-motion generation. Nonetheless, we value the suggestion and will ensure proper acknowledgment in the revised version.
> >
> > &nbsp;&nbsp;&nbsp;&nbsp;&nbsp;&nbsp;Thank you again for your valuable feedback and thoughtful consideration. If there are any remaining questions you would like to discuss further, we would be more than happy to provide any additional clarifications.

---

> > > ### Comment · Reviewer_oZAK · 2025-08-07
> > >
> > > No further questions. To summarize my opinion is that 1) the paper has some interesting insights but 2) the presentations is below NeurIPS standards -a) there are not enough figures b) there should be one unified video - not just an html file 3) the mesh based visualization which is claimed to be a benefit of the method should have been used for a majority of the visualizations which in the current html file is not 4) there are missing references which do not give enough credit to existing work. As such I will give a rating leaning towards rejection - though I won't be argueing very strongly for rejection - it might squeak in because of the other reviews

---

### Official Review · Reviewer_4XhZ · 2025-07-01

**Clarity:** 3
**Significance:** 3
**Originality:** 3
**Rating:** 5
**Confidence:** 5

**Summary:**

This paper introduces ACMDM, a motion diffusion model that uses absolute joint coordinates for text-to-motion generation. The model replaces the traditional kinematic-aware local-relative representations with simpler absolute coordinates, achieving high fidelity, better text alignment, and enhanced scalability. ACMDM also excels in downstream tasks like motion control and spatial editing, without requiring extensive re-engineering.

**Questions:**

I can imagine that the use of absolute coordinates might limit the diversity of the generated motions. Can the authors provide quantitative results on the diversity of the generated motions? It would be helpful to include both numerical data and visualizations to better illustrate this aspect. Additionally, how does the use of absolute coordinates affect the training costs? It would be valuable to understand how the computational and memory requirements change with model size and complexity.

**Ethical Concerns:**

["NO or VERY MINOR ethics concerns only"]

**Final Justification:**

The authors have addressed my concerns well. I now understand the rationale behind the diversity metrics, and the explanation makes sense—please consider incorporating this discussion into the paper. Computational efficiency also looks reasonable. Regarding the statement “We did not observe any noticeable artifacts on the HumanML3D test set,” I suggest testing more diverse text prompts beyond the dataset to better uncover potential failure cases. Overall, I find this method simple yet effective. I will keep my score and continue to support acceptance.

**Limitations:**

See the weakness part in the previous section.

**Paper Formatting Concerns:**

/

**Quality:**

3

**Strengths And Weaknesses:**

## Strengths

- The idea of using absolute coordinates for motion generation is simple yet effective. The results demonstrate high fidelity and better text alignment compared to previous methods.
- Nice Visualization: The paper provides well-made visualizations, making it easy to assess the model's performance :)
- The writing is good - this paper is easy to follow. Most related works are cited and discussed in this paper.
- I also like the experiment of scaling the models.

## Weaknesses

- Diversity concerns: While the approach improves text alignment, it's unclear whether it leads to a reduction in the diversity of generated motions. The absolute coordinate representation may limit variability.

- (minor) Including some failure cases would help readers better understand the model's limitations.

- (minor)There are some duplicated citations in the references (e.g., SCAMO appears twice...).

---

> ### Author Rebuttal · Authors · 2025-07-30
>
> # **To Reviewer 4XhZ:**
> &nbsp;&nbsp;&nbsp;&nbsp;&nbsp;&nbsp;Thank you for your insightful feedback and for recognizing the strength of our core idea, the quality of our visualizations, the clarity of our writing, and the design of our model scaling experiments. Below, we provide detailed responses to the weaknesses and questions you raised.
>
> ## **Q1: Response to Diversity Concerns.**
> &nbsp;&nbsp;&nbsp;&nbsp;&nbsp;&nbsp;We thank the reviewer for raising this important point. The MultiModality metric proposed by \[25\] and reported in Table 2 evaluates the diversity of generated motion embeddings conditioned on the same text prompt, as detailed in the Appendix. It is important to note that this is a secondary motion quality metric, meaning its value is only meaningful when core metrics such as FID and R-Precision are also strong.
> &nbsp;&nbsp;&nbsp;&nbsp;&nbsp;&nbsp;In our case, ACMDM achieves better FID and R-Precision compared to prior methods, while also maintaining strong MultiModality scores **(2.156 for ACMDM-S-PS2)**, even surpassing several relative-representation baselines such as **MLD++ \[14\] (1.924)** and **MotionLCM v2 \[14\] (1.780)**. This demonstrates that our absolute-coordinate formulation does not compromise motion diversity while maintaining better motion quality.
>
> ## **Q2: Response to Failure Cases.**
> &nbsp;&nbsp;&nbsp;&nbsp;&nbsp;&nbsp;We did not observe any noticeable artifacts on the HumanML3D \[25\] testset. However, similar to all baseline methods, for certain out-of-distribution prompts involving physically implausible scenarios such as "a person is flying", the model tends to produce motions like standing while waving hands, rather than attempting to achieve unrealistic flight behavior.
> &nbsp;&nbsp;&nbsp;&nbsp;&nbsp;&nbsp;Due to the NeurIPS rebuttal policy, we are unable to include additional video and image samples at this stage. However, we will provide extended qualitative results with failure case visualizations in the revised version.
>
> ## **Q3: Regarding Duplicate Citations.**
> &nbsp;&nbsp;&nbsp;&nbsp;&nbsp;&nbsp;We thank the reviewer for pointing this out. We will carefully review and correct this in the revised version.
>
> ## **Q4: Training and Computational Cost.**
> &nbsp;&nbsp;&nbsp;&nbsp;&nbsp;&nbsp;Thank you for the question. Our model uses a simple Transformer backbone, which is computationally similar to the baseline MDM, with the addition of AdaLN for conditioning. For the smallest variant, ACMDM-S-PS22 (8 layer 512 dim same as MDM \[91\]), training is computationally efficient, which only requires approximately **8 GB of GPU memory and completes in approximately 8 hours on RTX 4090**\. It is also faster to train than MDM, thanks to the temporal downsampling in our 2D-AutoEncoder. Full scaling details and configurations are provided in the Appendix. While the largest variant is more resource-intensive, it still trains within 2 days on an NVIDIA H200. Importantly, inference remains lightweight and can be performed on consumer GPUs such as the RTX 4090 or 5090\. ACMDM-S-PS22 achieves an **AITS (Average Inference Time per Sample) of 1.35s**. Moreover, since our diffusion objective is formulated as velocity, it supports the use of even fewer-step ODE solvers (e.g., ODE-10) during inference for further efficiency.

---

> > ### Comment · Reviewer_4XhZ · 2025-08-05
> >
> > The authors have addressed my concerns well.
> >
> > I now understand the current status and rationale behind the diversity metrics— and I agree with the authors' explanation, it would be better to incorporate the provided discussion in the revision. The computational efficiency also looks quite reasonable! As for the statement “We did not observe any noticeable artifacts on the HumanML3D [25] test set,” — I would suggest testing a broader range of text inputs that are similar to, but not directly from, the dataset to better identify potential failure cases. I like this simple yet effective method. I will keep my score and continue to support the acceptance of this paper.

---

> > > ### Author Response · Authors · 2025-08-05
> > >
> > > **Dear Reviewer 4XhZ:**
> > >
> > > &nbsp;&nbsp;&nbsp;&nbsp;&nbsp;&nbsp;Thank you very much for your thoughtful feedback and for engaging in the discussion. We are glad to hear that our responses have helped clarify most of your concerns, particularly regarding the diversity metrics and computational efficiency. We will also take your suggestion on testing a broader range of text inputs into account in the revised version. Thank you again for your feedback and continued recommendation for acceptance.

---

### Official Review · Reviewer_BjnS · 2025-07-01

**Clarity:** 2
**Significance:** 2
**Originality:** 2
**Rating:** 3
**Confidence:** 3

**Summary:**

This manuscript introduces ACMDM, a framework for human motion generation model that employs transformer diffusion models. The innovation of this work lies in that, compared with local-relative motion representation popularized by HumanML3D, the utilization of transformer improves text alignment, and strong scalability, even with a simple transformer backbone and no auxiliary kinematic-aware losses.

**Questions:**

1. In terms of the accuracy of the model, as shown in table 3 provided by the author, some indicators of the current method have reached the SOTA accuracy. It is hoped that the author can analyze the reasons why other accuracy is not significant and discuss whether there is a better solution?

2. It can be seen that the author's work is an improvement on HumanML3D, and local relative motion representation is not only in HumanML3D method. If possible, I hope the author can supplement more local motion representation methods and their advantages and disadvantages.

3. In section 3 of the manuscript, the author proposes the method of ACMDM model. Although it gave a good description of the implementation of the model, it lacked rigorous theoretical derivation and formula expression. If possible, I hope the author can supplement and improve the relevant theoretical research to demonstrate the effectiveness of the model.

**Ethical Concerns:**

["NO or VERY MINOR ethics concerns only"]

**Final Justification:**

The author has analyzed and supplemented the innovation and performance analysis, but the theoretical analysis only explains the reasons without supplementing its theory and formula. Therefore, I will maintain the original score

**Limitations:**

yes

**Quality:**

2

**Strengths And Weaknesses:**

Strengths:

1. The article proposes using a transformer approach to enhance the model's ability to handle absolute joint coordinates in the global space.
2. This paper clearly presents the problem and the corresponding solution.
3. The paper is easy to follow.


Weaknesses:

Incomplete performance analysis: Although SOTA results are achieved on most metrics, the paper lacks discussion on underperforming metrics and potential strategies for improvement.

Questionable novelty of ACMDM: The motivation of ACMDM module is to propose global motion representation for the lack of local relative motion representation, but this paper lacks the discussion on local relative motion representation.

Incomplete theory analysis: Although the implementation process of the model is described in Section 3, this paper lacks theoretical derivation and formula expression of the model.

---

> ### Author Rebuttal · Authors · 2025-07-30
>
> # **To Reviewer BjnS:**
> &nbsp;&nbsp;&nbsp;&nbsp;&nbsp;&nbsp;Thank you for your thoughtful feedback. We appreciate your recognition of the clarity with which we present the problem and solution, as well as the overall readability of the paper. Below, we provide detailed responses to the weaknesses and questions you raised.
>
> ## **Q1: Regarding Incomplete Discussion on Underperforming Metrics in Table 3 and Strategies for Improvement.**
> &nbsp;&nbsp;&nbsp;&nbsp;&nbsp;&nbsp;We appreciate the reviewer’s attention to detail and the opportunity to clarify. We would like to clarify that **ACMDM demonstrates SOTA performance across nearly all metrics in Table 3**, including FID, R-Precision Top-3, diversity, trajectory error, location error, and average error—**significantly outperforming baseline methods**. The only metric where ACMDM shows a slight shortfall is the foot skating ratio in the pelvis-only spatial control task, where it exceeds OmniControl \[102\] by just **0.0044 (0.44%)**. We note that this margin is minimal and within a very narrow threshold, particularly considering that the metric reflects the ratio of frames with foot movement beyond **2.5 cm** during contact. Furthermore, ACMDM still outperforms the next best baseline (GMD \[42\]) by a substantial margin of **0.0306 (3.06%)**, reinforcing its overall robustness.
>
> ## **Q2: Response to Questionable Novelty of ACMDM and Additional Relative Motion Representations.**
> &nbsp;&nbsp;&nbsp;&nbsp;&nbsp;&nbsp;We thank the reviewer for raising this point. We would like to clarify that the goal of our work is not to “propose a global motion representation to address the lack of local relative motion representations,” nor to introduce a novel model architecture. Rather, our motivation is to critically revisit the dominant practice in text-driven motion generation—the wide-spread adoption of root-relative (local) representations proposed by HumanML3D \[25\]—and demonstrate that a simple yet long-overlooked global (absolute) representation, when paired with a simple Transformer model, without task-specifc designs(*e.g.,* UNet or altered attentions) and any additional kinematic losses, can achieve significantly improved motion fidelity, better text alignment, strong scalability, and easy adaptability to subtasks such as motion control (Table 2\) and direct mesh generation (Section 3.3), without requiring task-specific reengineering.
> &nbsp;&nbsp;&nbsp;&nbsp;&nbsp;&nbsp;Notably, nearly all state-of-the-art baselines in both text-to-motion and motion control—such as MDM \[91\], MotionDiffuse \[120\], ReMoDiffuse \[119\], MLD++ \[14\], MotionLCM v2 \[14\], and MARDM \[66\]—rely on HumanML3D-style root-relative representations. In Table 1, we also include comparisons with the InterGen \[52\] representation (absolute \+ redundancy), and following the reviewer’s suggestion, we also trained ACMDM with SMPL \[59\] parameters—another relative representation but rarely used in text-to-motion generation—to provide a broader empirical perspective.
> &nbsp;&nbsp;&nbsp;&nbsp;&nbsp;&nbsp;As shown in Table A2, our approach significantly outperforms the SMPL parameter representation, also highlighting why such representations are rarely adopted in text-to-motion tasks. This further indicates that, despite its simplicity, our absolute-coordinate formulation is both highly effective and generalizable, with strong potential to serve as a foundation for future research in motion generation.
>
> &nbsp;&nbsp;&nbsp;&nbsp;&nbsp;&nbsp;**Table A2: Quantitative text-to-motion evaluation with SMPL parameter**
> | Methods | FID$\\downarrow$ | Top 1$\\uparrow$ | Top 2$\\uparrow$ | Top 3$\\uparrow$ | Matching$\\downarrow$ | CLIP-score$\\uparrow$ |
> | :---- | :---- | :---- | :---- | :---- | :---- | :---- |
> | ACMDM-S-SMPL | $2.765^{\\pm.436}$ | $0.303^{\\pm.028}$ | $0.461^{\\pm.012}$ | $0.609^{\\pm.020}$ | $4.233^{\\pm.028}$ | $0.557^{\\pm.005}$ |
> | ACMDM-S-Absolute Coordinate (**Ours**) | *$\mathbf{0.109}^{\\pm.005}$* | *$\mathbf{0.508}^{\\pm.002}$* | *$\mathbf{0.701}^{\\pm.003}$* | *$\mathbf{0.798}^{\\pm.003}$* | *$\mathbf{3.253}^{\\pm.010}$* | *$\mathbf{0.642}^{\\pm.001}$* |
>
>
> ## **Q3: Response to Incomplete Theoretical Derivation and Formulas in Section 3.**
> &nbsp;&nbsp;&nbsp;&nbsp;&nbsp;&nbsp;We thank the reviewer for the suggestion. We would like to clarify that our work does not propose a new model architecture or motion representation that necessitates new theoretical derivations. Instead, our contribution lies in critically re-examining the dominant practice in text-driven motion generation—the widespread use of root-relative (local) representations—and showing that a simple yet long-overlooked absolute-coordinate representation can be surprisingly effective. Even combining this representation with a simple Transformer architecture and no auxiliary kinematic losses, we achieve strong empirical performance across multiple tasks. We emphasize that our goal is not to introduce complex new formulations with theoretical derivation, but rather to challenge de facto assumptions through extensive experiments. Our motivation for this shift in representation is discussed in Lines 35–43. We believe the strength of our work lies in its empirical rigor and potential to reframe future research directions in this text-driven motion generation field.

---

> > ### Comment · Reviewer_suN9 · 2025-08-04
> >
> > I am interested in the concern raised by Reviewer BjnS about using a relative motion representation. In their response, the authors included experiments using the SMPL representation. While using it directly in the motion generation field isn't extremely common due to its learning difficulty, it's very useful as it preserves bone lengths. Additionally, almost all papers in the field convert their predictions to SMPL for visualization purposes, so being able to use SMPL directly would be quite beneficial.
> >
> > While the authors' experiments show that performance is significantly lower than when using absolute coordinates, I would like to ask how the pose rotations were represented in their experiments. The original SMPL representation uses axis-angles, but it is quite common to convert these rotations to 6D rotation matrices, as they provide an easier representation for data-driven approaches to learn.

---

> > > ### Author Response · Authors · 2025-08-05
> > > **Response to Reviewer suN9's Interest in Reviewer BjnS's Concern**
> > >
> > > # To Reviewer suN9:
> > >
> > > &nbsp;&nbsp;&nbsp;&nbsp;&nbsp;&nbsp;Thank you for your thoughtful question.
> > >
> > > &nbsp;&nbsp;&nbsp;&nbsp;&nbsp;&nbsp;In our original experiment, we used axis-angle rotations, following the original SMPL parameter representation. Based on your suggestion, we also conducted an additional experiment using 6D rotation form of SMPL parameter. The results, presented in Table B2 below show a slight improvement over axis-angle, but overall performance remains substantially below that of our absolute coordinate formulation. We believe one key reason is the **challenge of jointly modeling two distinct spaces—root translation and joint rotation**—a difficulty also reported in prior work [14]. Visually, this often leads to poor modeling of global movement (root transition), resulting in **unnatural root shifts and instability in global positioning**. Due to the NeurIPS rebuttal policy, we are unable to include additional video examples at this time, but we will include them in the updated visualizations post-review. This aligns with your observation—and our own experience—that SMPL parameter-based motion generation is **more difficult to learn**, despite its theoretical appeal.
> > >
> > > &nbsp;&nbsp;&nbsp;&nbsp;&nbsp;&nbsp;While it is true that SMPL preserves bone lengths by design, as we discussed in our rebuttal to your earlier question, **our absolute coordinate method also demonstrates strong bone length consistency**, both within generated sequences and when compared to ground truth. Additionally, our joint-level ACMDM outputs can also be converted into SMPL parameters for visualization purposes, just like the baseline methods, so the practical utility of SMPL-compatible visualization is retained.
> > >
> > > &nbsp;&nbsp;&nbsp;&nbsp;&nbsp;&nbsp;Ultimately, we believe this highlights an important distinction between **what is expected to be beneficial**—such as bone-preserving, industry-similar formats like SMPL—and **what is practically effective and learnable** within a generative modeling framework. Our results suggest that while SMPL offers theoretical advantages, it remains challenging to model effectively in practice. In contrast, absolute coordinates, by being fully global and task-agnostic, enable high-quality motion generation while also simplifying downstream applications such as motion control and direct mesh generation—without requiring task-specific re-engineering.
> > >
> > > **Table B2: Quantitative text-to-motion evaluation with SMPL parameter 6D**
> > >
> > > | Methods | FID$\\downarrow$ | Top 1$\\uparrow$ | Top 2$\\uparrow$ | Top 3$\\uparrow$ | Matching$\\downarrow$ | CLIP-score$\\uparrow$ |
> > > | :---- | :---- | :---- | :---- | :---- | :---- | :---- |
> > > | ACMDM-S-PS22-SMPL-6D | $2.037^{\\pm.328}$ | $0.339^{\\pm.017}$ | $0.513^{\\pm.010}$ | $0.664^{\\pm.009}$ | $3.936^{\\pm.026}$ | $0.565^{\\pm.007}$ |

---

> > ### Author Response · Authors · 2025-08-07
> >
> > **Dear Reviewer BjnS:**
> >
> > &nbsp;&nbsp;&nbsp;&nbsp;&nbsp;&nbsp;We would like to sincerely thank you again for your valuable feedback and engagement in the discussion. As the rebuttal period draws to a close, we would like to respectfully inquire whether any concern remains, particularly in light of our most recent response.
> >
> > &nbsp;&nbsp;&nbsp;&nbsp;&nbsp;&nbsp;We would be grateful for any additional comments you may have, and we are more than happy to provide further clarifications. Thank you again for your time and thoughtful consideration

---

> > > ### Comment · Reviewer_BjnS · 2025-08-08
> > >
> > > Thank you for your detailed response. I think current comment has not fully addressed my concern.
> > >
> > > Although the author added an explanation, my concern is the same as that of the reviewer oZAK. Therefore, I plan to maintain my current rating of Borderline Reject for now and will make a final decision after further discussion with the other reviewers.

---

> > ### Author Response · Authors · 2025-08-08
> > **Response to Reviewer BjnS's New Comments**
> >
> > ## **Dear Reviewer BjnS:**
> >
> > &nbsp;&nbsp;&nbsp;&nbsp;&nbsp;&nbsp;Thank you very much for your response and for taking the time to engage with our rebuttal.
> >
> > &nbsp;&nbsp;&nbsp;&nbsp;&nbsp;&nbsp;May we respectfully ask if any concerns from your original review remain unaddressed? We would be happy to provide additional clarification before the discussion deadline.
> >
> > &nbsp;&nbsp;&nbsp;&nbsp;&nbsp;&nbsp;Since you referenced Reviewer oZAK’s concerns as your remaining concern, we would sincerely like to offer the following clarifications.
> >
> > ### **On concerns of visualization formats:**
> > > there should be one unified video - not just an html file
> >
> > &nbsp;&nbsp;&nbsp;&nbsp;&nbsp;&nbsp;We explicitly chose the HTML format to **enable readers to interactively browse extensive results by task** (e.g., text-to-motion, motion control, direct mesh generation) **without the need to sit through a long video**. We chose this format to allow faster navigation, side-by-side comparisons, and broader qualitative coverage, which are often not feasible with a traditional one linear video. This is also a common practice in prior works, such as MoMask [23].
> >
> > > the mesh based visualization which is claimed to be a benefit of the method should have been used for a majority of the visualizations which in the current html file is not
> >
> > &nbsp;&nbsp;&nbsp;&nbsp;&nbsp;&nbsp;We would like to respectfully clarify that direct mesh generation is only one of the three tasks presented as contributions in our work, and accordingly, mesh-based visualizations are used consistently within that context. For joint-level tasks (generation and control), we follow the skeletal task design defined in [25] and visualize joint skeletons, as all baseline methods directly output joints, not mesh. This is also a common practice in prior works (e.g., MotionDiffuse [119] and MARDM [14]). For mesh-level tasks, we provide extensive mesh-based visualizations to highlight our method’s potential for direct mesh generation. This distinction allows us to respect task-specific conventions while showcasing our method’s flexibility.
> >
> > > there are not enough figures
> >
> > &nbsp;&nbsp;&nbsp;&nbsp;&nbsp;&nbsp;We believe the **temporal and 4D nature of motion is best conveyed through video**, not as static figure. In the main paper, we include two key visualizations in the main paper (a teaser and a side-by-side comparison). To complement this, we also provide an extensive set of results in the supplementary HTML webpage, covering over 45 motion clips across three distinct tasks. We sincerely hope the reviewers will consider the full scope of visualizations included in the video results as part of the overall evaluation.
> >
> > &nbsp;&nbsp;&nbsp;&nbsp;&nbsp;&nbsp;Respectively, we would also like to note that **other reviewers (e.g., suN9 and 4XhZ) have also explicitly praised the clarity and quality of our visualizations**.
> >
> > ### **On MOJO citation:**
> >
> > &nbsp;&nbsp;&nbsp;&nbsp;&nbsp;&nbsp;As mentioned in our rebuttal, we will cite MOJO in the revised version of the paper. However, MOJO (CVPR 2021) predates HumanML3D [25] (CVPR 2022), and **primarily focuses on a different task**: human motion prediction. They adopt surface keypoints as their representation, rather than joint coordinates or full mesh vertices. This aligns with our statement in the introduction section that the field largely converged on root-relative joint representations following the introduction of HumanML3D. Our framing of absolute coordinates as “simple and long-abandoned” reflects this trajectory. We do not claim the representation is novel, but rather that it has been overlooked for text-to-motion in recent years and deserves rethinking and empirical re-examination
> >
> > &nbsp;&nbsp;&nbsp;&nbsp;&nbsp;&nbsp;Thank you again for your valuable feedback and thoughtful consideration. If there are any remaining questions you would like to discuss further, we would be more than happy to provide any additional clarifications before the deadline.

---

> > > ### Comment · Reviewer_BjnS · 2025-08-09
> > >
> > > Thank you for your detailed response. I think current comment has not fully addressed my concern.
> > >
> > > In my view, most of my concerns have been adequately addressed. However, I remain skeptical about the theoretical framework's supplementary components. The author's previous responses primarily focused on explaining the model's significance and motivations, without establishing formalized derivations. Section 3 of the manuscript appears more akin to a technical report than an academic paper.

---

> ### Comment · Reviewer_BjnS · 2025-08-04
>
> Thank you for your detailed response. I think current rebuttal has not fully addressed my concern.
>
> For Q3, although it has been experimentally demonstrated that its method works, it does not explain why it works from a theoretical perspective? In the new data application scenario, the model will have different feature representations, which I think is also a necessary part of the explanation.

---

> > ### Author Response · Authors · 2025-08-05
> > **Response to Reviewer BjnS’s Followup Comments**
> >
> > # To Reviewer BjnS:
> > &nbsp;&nbsp;&nbsp;&nbsp;&nbsp;&nbsp;Thank you for engaging with our rebuttal and providing follow-up comments. Below, we provide detailed responses to the followup questions you raised.
> >
> > &nbsp;&nbsp;&nbsp;&nbsp;&nbsp;&nbsp;We appreciate the emphasis on understanding why our method works. We would like to respectfully clarify that formal theoretical explanations for representation learning in deep generative models remain an open research problem. As seen in many prior influential works (e.g., MAE [1], REPA [2]), it is common for contributions to be supported primarily by empirical evidence rather than formal proofs, and empirical advances are ahead of formal proofs.
> >
> > &nbsp;&nbsp;&nbsp;&nbsp;&nbsp;&nbsp;Nevertheless, we are happy to share the underlying theoretical hypothesis, which aligns with our motivation stated in Lines 35–43 of our paper:
> >
> > &nbsp;&nbsp;&nbsp;&nbsp;&nbsp;&nbsp;(1) **Operating in absolute XYZ coordinates provides inherent spatial awareness**, enabling the model to naturally align with human instructions or prompts, which are often defined in global terms (e.g., “raise hand to **1.2m**”). This representation also makes it straightforward to incorporate spatial constraints (e.g., in Section 3.2, controllable motion generation which primarily follows absolute position targets). In contrast, relative-to-root representations lack the acknowledgment of the absolute spatial positions and constraints in global space.
> >
> > &nbsp;&nbsp;&nbsp;&nbsp;&nbsp;&nbsp;(2) **Absolute coordinates enable more meaningful contact reasoning**. Contact can be directly modeled as point-to-point interactions in global 3D (XYZ) space. Relative representations, by design, obscure global location and make such spatial relationships much harder to capture.
> >
> > &nbsp;&nbsp;&nbsp;&nbsp;&nbsp;&nbsp;(3) **Absolute coordinate representation promotes greater generality across tasks and datasets**, as it is not tied to any dataset-specific representation design. Since each point (e.g., joint, mesh vertex) is simply represented by its (x, y, z) location in space, the same representation can be used seamlessly for joints, meshes, etc—supporting broader generalization and easier adaptation to downstream tasks.
> >
> > &nbsp;&nbsp;&nbsp;&nbsp;&nbsp;&nbsp;Our experiments across three diverse settings: HumanML3D (22 joints), KIT-ML (in Appendix; 21 joints), and mesh-level HumanML3D (6890 vertices), provide strong empirical support for this hypothesis and highlight the method’s robustness under varying scenarios.
> >
> > &nbsp;&nbsp;&nbsp;&nbsp;&nbsp;&nbsp; At the same time, we are confused about your reference to "new data application scenarios" with "different feature representations." We will use our best guess to address it below. Any of your further clarifications would be appreciated.
> >
> > &nbsp;&nbsp;&nbsp;&nbsp;&nbsp;&nbsp;Our method is specifically **designed to avoid dataset- or task-specific encodings**. Unlike root-relative representations (e.g., those used in HumanML3D), which require customized preprocessing and vary in dimensionality across datasets (e.g., 263D for HumanML3D, 251D for KIT), our approach uses a universal representation where every joint or mesh vertex is simply a point in 3D space (XYZ). This absolute coordinate formulation is minimal, consistent, and agnostic to the number of points, whether it’s 22 joints, 21 joints, or 6890 mesh vertices, as supported in our experiments. The model sees each as just a collection of 3D points where no special structure, design, or root-relative transformation is needed. This makes our method broadly applicable across datasets and tasks, without requiring task-specific reengineering.
> >
> > &nbsp;&nbsp;&nbsp;&nbsp;&nbsp;&nbsp;We thank you again for your thoughtful feedback.
> >
> > [1] He, Kaiming, et al. "Masked autoencoders are scalable vision learners." Proceedings of the IEEE/CVF conference on computer vision and pattern recognition. 2022.
> >
> > [2] Yu, Sihyun, et al. "Representation alignment for generation: Training diffusion transformers is easier than you think." arXiv preprint arXiv:2410.06940 (2024).

---

### Official Review · Reviewer_suN9 · 2025-07-03

**Clarity:** 3
**Significance:** 3
**Originality:** 3
**Rating:** 5
**Confidence:** 4

**Summary:**

In this work, the authors propose using only global joint positions to represent human motions and as the input representation for training generative models. To validate their approach against larger and more established representations that include redundancy and relative information, they conduct an in-depth analysis of several design choices. Their findings demonstrate that this representation is more suitable for a wide range of applications. Finally, for representing motions as meshes, they propose a method based on direct mesh prediction instead of traditional approaches that rely on fitting to parametric models.

**Questions:**

Based on the weaknesses identified in the previous section, I have the following questions and suggestions:

1. The paper adopts a new, reduced motion representation for evaluation. Were all baseline models retrained using this representation to ensure a fair comparison? If not, could the authors provide results using the original HumanML3D format and standard evaluation metrics? This would help clarify how much of the performance gain is attributable to the new representation versus the proposed model itself.
2. Since the method relies on global joint positions without incorporating kinematic losses, how is bone length consistency maintained? Are there any failure cases, particularly with out-of-distribution motions? Providing quantitative or visual evidence on this aspect would strengthen the validity of the approach.
3. While the mesh-based output is promising, its practical usability in real-world applications remains unclear. How does the proposed approach integrate with standard rigging pipelines that depend on joint rotations and skinning techniques (e.g., linear blend skinning)? Additionally, what is the computational performance of the direct mesh generation method (e.g., inference time per motion sample)?
4. Although the paper presents strong qualitative results, a user study would offer a more systematic and objective assessment of perceptual quality. Could the authors consider including such an evaluation?

If these points are adequately addressed, I would be inclined to increase my overall score for this submission.

**Ethical Concerns:**

["NO or VERY MINOR ethics concerns only"]

**Final Justification:**

Based on the authors' rebuttal and the subsequent discussion, I've decided to raise my original rating to Accept.

**Limitations:**

Yes.

**Quality:**

3

**Strengths And Weaknesses:**

In general, I really liked this work, and I believe it has several strengths that are worth highlighting:

- The quality and clarity of the writing are excellent. The paper is well written, easy to follow, and clearly communicates the ideas.
- The analysis and ablations presented are thorough and do not leave any major gaps. The experiments address key design choices commonly made in the development of motion diffusion models.
- The supplementary material is extensive and includes all the necessary details to fully understand the work. It also provides a large number of visualizations that help qualitatively assess the quality of the proposed method.

However, there are some weaknesses and questions I would like the authors to clarify:

- As mentioned in Appendix B, the evaluation framework used in this paper is a newly introduced, more robust version that relies on a reduced representation including only fundamental motion components. This raises a concern: were all the baseline methods retrained using this new reduced representation to ensure a fair comparison? If so, is it possible to adapt the proposed representation to the original HumanML3D format and evaluate it using the standard metrics and evaluators? Since all baseline methods were originally designed for the HumanML3D representation, this comparison would be valuable and could offer additional insights.
- One of the challenges of using global joint positions without kinematic losses is ensuring consistent bone lengths. What happens when the model generates highly out-of-distribution motions? How is bone length consistency maintained with this representation?
- Regarding the direct mesh generation approach, while I found the results promising, I have concerns about the practical usability of this representation. In real-world applications like video games or animation, motion data is typically applied to rigged meshes, which benefit from relative joint rotations and techniques like linear blend skinning for efficient animation. What is the actual performance and generalizability of the direct mesh generation method? How do the authors envision using this representation in such application scenarios where rigged, animation-ready meshes are the norm?
- Finally, while the paper provides a large number of qualitative examples, it would have been beneficial to include a user study to assess qualitative quality in a more systematic and reliable way.

---

> ### Author Rebuttal · Authors · 2025-07-30
>
> # **To Reviewer suN9:**
> &nbsp;&nbsp;&nbsp;&nbsp;&nbsp;&nbsp;Thank you for your insightful feedback and your recognition of our work’s clarity in writing, thoroughness in analysis, and the quality and quantity of our visual results. Below, we provide detailed responses to the weaknesses and questions you raised.
>
> ## **Q1: Response to Fairer Comparison with Baseline Methods.**
> &nbsp;&nbsp;&nbsp;&nbsp;&nbsp;&nbsp;Following the reviewer’s suggestion, and to ensure a fairer comparison, we retrained all baseline methods that originally used the full HumanML3D representation with the reduced, essential representation and present the results in Table A1. While most baselines showed only minor performance fluctuations, MLD++ \[14\] and MotionLCM v2 \[14\] exhibited more noticeable changes on the FID metric. This suggests that these methods may overfit to redundant dimensions when trained on the original HumanML3D \[25\] representation. Nevertheless, **our method consistently achieves state-of-the-art performance** across major metrics.  We would also like to emphasize that, unlike baseline methods, our approach is not only effective for text-to-motion generation but also well-suited for downstream tasks and demonstrates strong scalability.
> &nbsp;&nbsp;&nbsp;&nbsp;&nbsp;&nbsp;As highlighted in MARDM \[66\], the original HumanML3D evaluator is problematic because (1) it unnecessarily emphasizes redundant dimensions, and (2) all baseline methods can directly output essential, animatable dimensions without any further conversion. We thus strongly argue against evaluating methods with redundant dimensions, as it can be both inaccurate and misleading for real animation quality. Importantly, we believe that all motion generation methods should be optimized for producing the most perceptually faithful outputs, rather than fitting to evaluator-specific artifacts. Therefore, we also include results from a user study (see Response 4\) to provide a more reliable and human-aligned assessment of generated motion quality.
>
> &nbsp;&nbsp;&nbsp;&nbsp;&nbsp;&nbsp;**Table A1: Quantitative text-to-motion evaluation with baseline methods retrained with reduced representation.**
> | Methods | FID$\\downarrow$ | Top 1$\\uparrow$ | Top 2$\\uparrow$ | Top 3$\\uparrow$ | Matching$\\downarrow$ | MModality$\\uparrow$ | CLIP-score$\\uparrow$ |
> | :---- | :---- | :---- | :---- | :---- | :---- | :---- | :---- |
> | MDM-50Step-Reduced \[91\]  | $0.491^{\\pm.027}$ | $0.435^{\\pm.006}$ | $0.631^{\\pm.006}$ | $0.733^{\\pm.005}$ | $3.659^{\\pm.031}$ | **$\mathbf{3.627}^{\\pm.033}$** | $0.571^{\\pm.003}$ |
> | MotionDiffuse-Reduced \[119\]  | $0.535^{\\pm.010}$ | $0.459^{\\pm.005}$ | $0.651^{\\pm.004}$ | $0.758^{\\pm.004}$ | $3.481^{\\pm.021}$ | *$3.158^{\\pm.041}$* | $0.610^{\\pm.005}$ |
> | ReMoDiffuse-Reduced \[120\]  | $0.857^{\\pm.017}$ | $0.463^{\\pm.004}$ | $0.648^{\\pm.004}$ | $0.744^{\\pm.005}$ | $3.423^{\\pm.021}$ | $2.810^{\\pm.157}$ | $0.618^{\\pm.003}$ |
> | MLD++-Reduced \[14\]  | $0.181^{\\pm.005}$ | $0.506^{\\pm.003}$ | $0.697^{\\pm.002}$ | $0.796^{\\pm.001}$ | $3.257^{\\pm.009}$ | $1.879^{\\pm.059}$ | $0.641^{\\pm.002}$ |
> | MotionLCM V2-Reduced \[14\]  | $0.203^{\\pm.007}$ | $0.503^{\\pm.002}$ | $0.696^{\\pm.002}$ | $0.794^{\\pm.002}$ | $3.260^{\\pm.010}$ | $1.795^{\\pm.077}$ | $0.640^{\\pm.001}$ |
> | MARDM-$\\boldsymbol{\\epsilon}$-Originally Reduced \[66\] | $0.116^{\\pm.004}$ | $0.492^{\\pm.006}$ | $0.690^{\\pm.005}$ | $0.790^{\\pm.005}$ | $3.349^{\\pm.010}$ | $2.470^{\\pm.053}$ | $0.637^{\\pm.005}$ |
> | MARDM-$\\mathbf{v}$-Originally Reduced \[66\] | $0.114^{\\pm.007}$ | $0.500^{\\pm.004}$ | $0.695^{\\pm.003}$ | $0.795^{\\pm.003}$ | $3.270^{\\pm.009}$ | $2.231^{\\pm.071}$ | $0.642^{\\pm.002}$ |
> | ACMDM-S-PS22 | *$0.109^{\\pm.005}$* | *$0.508^{\\pm.002}$* | *$0.701^{\\pm.003}$* | *$0.798^{\\pm.003}$* | *$3.253^{\\pm.010}$* | $2.156^{\\pm.061}$ | *$0.642^{\\pm.001}$* |
> | ACMDM-XL-PS2 | **$\mathbf{0.058}^{\\pm.004}$** | **$\mathbf{0.522}^{\\pm.002}$** | **$\mathbf{0.713}^{\\pm.002}$** | **$\mathbf{0.807}^{\\pm.002}$** | **$\mathbf{3.205}^{\\pm.008}$** | $2.077^{\\pm.083}$ | **$\mathbf{0.652}^{\\pm.001}$** |
>
>
>
>
> ## **Q2: Response to Bone Length Consistency with Absolute Joint Coordinates.**
> &nbsp;&nbsp;&nbsp;&nbsp;&nbsp;&nbsp;We thank the reviewer for raising this point. We would like to clarify a common misconception: relative joint positions do not inherently guarantee constant bone lengths. It is possible to predict arbitrarily small or large relative offsets that violate bone length constraints. Only rotational representations (e.g., joint rotations) inherently preserve bone lengths by using a fixed template skeleton. However, the rotation data in HumanML3D is known to be redundant and inaccurate (as discussed in our paper Line 28, and noted in HumanML3D’s GitHub issue), thus unable to be used for animation. Consequently, no baseline method using HumanML3D’s relative joint positions can strictly preserve bone length.
> &nbsp;&nbsp;&nbsp;&nbsp;&nbsp;&nbsp;To evaluate ACMDM’s ability to maintain bone length consistency, we conducted two quantitative tests on the HumanML3D testset:
> &nbsp;&nbsp;&nbsp;&nbsp;&nbsp;&nbsp;(1) Intra-sequence bone length stability – measuring changes in bone lengths within a generated sequence.
> &nbsp;&nbsp;&nbsp;&nbsp;&nbsp;&nbsp;(2) Bone length consistency with ground truth – comparing generated bone lengths against those from the ground truth.
> &nbsp;&nbsp;&nbsp;&nbsp;&nbsp;&nbsp;We also compared ACMDM to SOTA relative representation baselines with results presented in Table B1. Results indicate that **ACMDM not only maintains stable bone lengths but also outperforms some relative baselines**, such as MotionLCM V2 and MARDM.
> &nbsp;&nbsp;&nbsp;&nbsp;&nbsp;&nbsp;Due to the NeurIPS rebuttal policy, we cannot include additional videos, but we observed no noticeable bone length artifacts in the HumanML3D testset. To further probe robustness, we tested ACMDM on 10 out-of-distribution prompts involving “flying” (e.g., “a person is flying in the air”), which are impossible motions. Despite the unusual prompts, the generated motions mostly showed realistic behavior (e.g., standing, waving), and intra-sequence bone length variation remained very low (0.0071 m), confirming the reliability of our approach.
>
> &nbsp;&nbsp;&nbsp;&nbsp;&nbsp;&nbsp;**Table B1: Bone length consistency on the humanml3d testset.**
> | Method | Intra-Seq Variation (m) $\\downarrow$ | *v.s.* GT Bone Length (m) $\\downarrow$ |
> | :---- | :---- | :---- |
> | MLD++ \[14\] | 0.0053 | 0.0057 |
> | MotionLCM V2 \[14\] | 0.0077 | 0.0073 |
> | MARDM-v \[66\] | 0.0070 | 0.0071 |
> | ACMDM-S-PS22 | 0.0061 | 0.0069 |
> | ACMDM-XL-PS2 | 0.0059 | 0.0065 |
>
>
>
>
> ## **Q3: Regarding Future Envision and Practical Usability of Direct Mesh Generation.**
> &nbsp;&nbsp;&nbsp;&nbsp;&nbsp;&nbsp;In the paper, we demonstrate strong results with direct SMPL-H \[59\] mesh generation. We view direct mesh generation as the next step toward simplifying the 3D character animation pipeline. Historically, animators had to draw each frame of a character’s motion manually. The introduction of rigging and skinning revolutionized this process by enabling skeleton-based animation to drive meshes, significantly reducing human labor. Our approach aims to take this one step further, providing an end-to-end solution where a user only needs to provide a text prompt or a static 3D character, and the generative model directly produces realistic animated motion without intermediate skeletons or rigging.
> &nbsp;&nbsp;&nbsp;&nbsp;&nbsp;&nbsp;As highlighted in our visualizations, direct mesh modeling captures finer 3D details and implicit motion dynamics that skeleton-to-mesh pipelines either fail to reproduce or require additional manual work to achieve. Moreover, by representing all characters as sets of absolute coordinates, the absolute-coordinate approach has the potential to generalize across diverse characters without the need for predefined kinematic chains for each character.
> &nbsp;&nbsp;&nbsp;&nbsp;&nbsp;&nbsp;We believe that, in the future, this direction will offer a more universal, streamlined, and automation-friendly solution for character animation, reducing manual labor while achieving higher motion fidelity.
> &nbsp;&nbsp;&nbsp;&nbsp;&nbsp;&nbsp;Regarding computational resources, as discussed in Section 3.3, our diffusion model operates in latent space by encoding SMPL-H meshes (6890×3) into compact latent representations of size (28×12). With motion diffusion patch size set to 1x28, the inference cost becomes equivalent to that of joint-level generation (for smallest variant ACMDM-S-PS28, **AITS=2.37s**), making our approach very efficient despite working directly with mesh data.
>
>
> ## **Q4: User Study.**
> &nbsp;&nbsp;&nbsp;&nbsp;&nbsp;&nbsp;We conducted a user study with 15 participants to evaluate ACMDM against SOTA baseline methods across a total of 21 motion comparisons, including both Text-to-Motion generation (15 comparisons) and Motion Control tasks (6 comparisons), with motion captions drawn from the HumanML3D test set. For each comparison, participants were shown side-by-side outputs from ACMDM and baseline methods and asked to choose the motion that appeared most natural, text-aligned, and, on Motion Control, also accurately followed the control signal.
> &nbsp;&nbsp;&nbsp;&nbsp;&nbsp;&nbsp;In Text-to-Motion generation, **ACMDM was the most preferred method**, selected in **62.22% (140/total 225\)** of all cases, compared to 15.56% (35/225) for MLD++, 8.44% (19/225) for MotionLCM v2, and 13.78% (31/225) for MARDM. In Motion Control, **ACMDM was again the most preferred**, with **78.89% (71/total 90)**, outperforming OmniControl (21.11%; 19/90 ) and MotionLCM v2 \+ CtrlNet (0% 0/90). These results demonstrate that ACMDM is consistently favored by users in terms of perceptual quality and control fidelity.

---

> > ### Comment · Reviewer_suN9 · 2025-08-04
> > **Follow-up on Direct Mesh Generation (Q3)**
> >
> > I would like to thank the authors for the detailed answers in their rebuttal as they have helped clarify some of my doubts. However, I still have concerns about Direct Mesh Generation (Q3).
> >
> > While the efficiency shown by the authors' approach is great, I am concerned about how this approach can scale. The SMPL template mesh has a relatively simple geometry, but the proposed method would be quite challenging to implement with complex, real-world meshes, such as those used in the video game and film industries. It's also important to consider the complex systems used in those industries, such as cloth simulations. Additionally, for these specific cases, it would be necessary either to retrain a model for each different mesh or to train a very complex model capable of handling different mesh topologies. Finally, if a generated motion needed to be applied to a different mesh using a technique like retargeting, the proposed method would require estimating joint positions from the predicted mesh before the retargeting could be performed.
> >
> > Wouldn't it be easier to directly predict a skeleton-based relative rotation representation, which is an industry standard and is more flexible for minor tweaks by artists and traditional animation techniques? Can the authors provide their views on these limitations?

---

> > > ### Author Response · Authors · 2025-08-05
> > > **Response to Reviewer suN9's Followup Comments**
> > >
> > > # To Reviewer suN9:
> > >
> > > &nbsp;&nbsp;&nbsp;&nbsp;&nbsp;&nbsp;Thank you for your thoughtful response. Below, we provide detailed responses to the followup questions you raised.
> > >
> > > &nbsp;&nbsp;&nbsp;&nbsp;&nbsp;&nbsp;We would like to respectfully clarify that the main contribution of our work is to **revisit and advocate for a simple yet effective representation—absolute coordinates**—which simplifies motion generation, including both joint-level and controllable motion generation. The ability to generate motion at the mesh level naturally generalizes from this formulation, as mesh vertices can be considered a subgroup of absolute coordinates. We present direct mesh generation as a natural generalization of our framework and a promising future direction, offering a possible alternative perspective to traditional skinning+rigging methods (whose representations are typically hard to model by generative models).
> > >
> > > &nbsp;&nbsp;&nbsp;&nbsp;&nbsp;&nbsp;We kindly note that delivering a fully operational, industry-grade system is **beyond the intended scope of this paper**. Our intention is not to immediately replace current production pipelines, but rather to encourage further research and exploration of this direction as an emerging and potential approach. Nevertheless, we are glad to discuss the broader potential of this direction:
> > >
> > > &nbsp;&nbsp;&nbsp;&nbsp;&nbsp;&nbsp;**Modeling complex mesh motions**. We believe that with a strong AutoEncoder or VAE—like those commonly used in 3D generation pipelines—it is indeed feasible to compress complex meshes into a latent representation. Once in this form, the generation process becomes straightforward and can follow the same formulation as presented in our paper.
> > >
> > > &nbsp;&nbsp;&nbsp;&nbsp;&nbsp;&nbsp;**Cloth simulation**. We would like to respectfully point out that traditional pipelines do not handle this directly either. Instead, they typically rely on additional manual effort and specialized post-processing tools applied after rigging and skinning to simulate dynamic mesh and garment behavior. In contrast, our formulation shows early potential to capture finer 3D details and implicit motion dynamics directly, as demonstrated in our visualizations, suggesting that end-to-end modeling of such dynamics may be possible.
> > >
> > > &nbsp;&nbsp;&nbsp;&nbsp;&nbsp;&nbsp;**Modeling different mesh and shapes**. It’s important to note that our formulation does not necessarily require training a separate model for each mesh or shape. In contrast, skeleton-based pipelines often require mesh-specific retargeting logic or re-parameterization of rotation targets. While a complete application to highly diverse meshes and shapes is beyond the scope of this paper, we provide preliminary evidence pointing toward the feasibility of this direction. Our method can condition on a 3D T-Pose SMPL-H mesh of different shapes and generate shape-aware motion. As shown in Table C1, we finetuned an ACMDM-Mesh-S-PS28 with ControlNet that takes a 3D T-pose SMPL-H mesh as a condition. The results indicate no degradation in generation quality and the capability of following the bone lengths (v.s. GT Bone Length) and overall shape (LSD) of the provided mesh.
> > >
> > > &nbsp;&nbsp;&nbsp;&nbsp;&nbsp;&nbsp;Finally, while we fully acknowledge the practicality of root-translation+joint-angle pipelines widely used in industry, we would like to reiterate—as also noted in your response to Reviewer BjnS—that such representations are typically more difficult to model and lack global spatial awareness due to their relative nature. Our work is not intended to oppose or immediately replace these established standards, but rather to offer a different perspective that may inspire new ways of thinking about motion representation and the design of future animation pipelines.
> > >
> > > &nbsp;&nbsp;&nbsp;&nbsp;&nbsp;&nbsp;We thank you again for your thoughtful feedback.
> > >
> > > **Table C1: Preliminary results on direct SMPL-H mesh generation conditioned on different 3D T-pose meshes.**
> > >
> > > | Methods | FID$\\downarrow$ | Top 1$\\uparrow$ | Top 2$\\uparrow$ | Top 3$\\uparrow$ | Matching$\\downarrow$ | CLIP-score$\\uparrow$ | LSD$\\downarrow$ | Intra-Seq Variation (m)$\\downarrow$  | v.s. GT Bone Length (m) $\\downarrow$ |
> > > | :---- | :---- | :---- | :---- | :---- | :---- | :---- | :---- | :---- | :---- |
> > > | ACMDM-Mesh-S-PS28 | $0.211^{\\pm.005}$ | $0.478^{\\pm.004}$ | $0.682^{\\pm.003}$ | $0.784^{\\pm.003}$ | $3.405^{\\pm.011}$ | $0.620^{\\pm.002}$ | $0.0026^{\\pm.0002}$ | $0.0069^{\\pm.0003}$ | $0.0078^{\\pm.0004}$ |
> > > | ACMDM-Mesh-S-PS28-CtrlNet-All-Shapes | $0.197^{\\pm.007}$ | $0.485^{\\pm.003}$ | $0.689^{\\pm.003}$ | $0.790^{\\pm.002}$ | $3.384^{\\pm.014}$ | $0.623^{\\pm.003}$ | $0.0029^{\\pm.0003}$ | $0.0073^{\\pm.0003}$ | $0.0077^{\\pm.0005}$ |

---

### Decision · Program_Chairs · 2025-09-17

**Decision:**

Reject

**Comment:**

This paper introduces a motion diffusion model using global joint positions for text-to-motion generation. The reviewers highlight several strengths of this work, such as that the idea of using absolute coordinates is simple yet effective, that most design choices are verified through experiments, and that the practical value of this work might be significant.

After the rebuttal and the reviewer-author discussion, two reviewers (BjnS, oZAK) still have negative concerns. Specifically, reviewer BjnS argued that the theoretical derivation is missing. Although a rigorous theoretical proof might not be suitable for the task that this paper tackles, a better presentation of Section 3 with more analysis of the motivation or insights might be helpful to readers in better understanding the reasons for the design choices. This comment is somewhat similar to what reviewer oZAK pointed out: that the presentation of this work could be improved. Reviewer oZAK also mentioned that a unified demo video and mesh-based visualization should be provided. Additionally, there is not enough discussion of the related works.

Overall, though the strengths of this work are obvious, the current version of the paper can still be improved significantly. This paper is recommended for rejection, and the authors are encouraged to incorporate the feedback from the reviewers and try another venue.